

# The EarthCARE Mission - Science and System Overview

Tobias Wehr[1,†], Takuji Kubota[2], Georgios Tzeremes[1], Kotska Wallace[1], Hirotaka Nakatsuka[2],
Yuichi Ohno[4], Rob Koopman[1], Stephanie Rusli[1], Maki Kikuchi[2], Michael Eisinger[3], Toshiyuki Tanaka[2],
Masatoshi Taga[2], Patrick Deghaye[1], Eichi Tomita[2], and Dirk Bernaerts[1]

[1]European Space Agency, ESA-ESTEC, Keplerlaan 1, 2201 AZ Noordwijk, The Netherlands
[2]Japan Aerospace Exploration Agency (JAXA) Japan, 305-8505 2 Chome-1-1, Sengen, Tsukuba, Ibaraki, Japan
[3]European Space Agency, ESA-ECSAT, Fermi Avenue, Didcot OX11 0FD, United Kingdom
[4]National Institute of Information and Communications Technology (NICT) Japan, 184-8795 4-2-1 Nukui-Kitamachi,
Koganei, Tokyo, Japan
[†]deceased, 1 February 2023

**Correspondence:** Kotska Wallace (kotska.wallace@esa.int), Takuji Kubota (kubota.takuji@jaxa.jp)

**Abstract.** The Earth Cloud Aerosol and Radiation Explorer (EarthCARE) is a satellite mission implemented by the European Space Agency (ESA) in cooperation with the Japan Aerospace Exploration Agency (JAXA) to measure global profiles of aerosols, clouds and precipitation properties together with radiative fluxes and derived heating rates. The data will be used in particular to evaluate the representation of clouds, aerosols, precipitation and associated radiative fluxes in weather forecasting

and climate models.

The satellite scientific payload consists of four instruments, a lidar, a radar, an imager and a broad-band radiometer. The measurements of these instruments are processed in the ground segment, which produces and distributes the science data products.

The EarthCARE observational requirements are addressed. An overview is given of the space segment with a detailed
description of the four science instruments. Furthermore, the elements of the Space Segment and Ground Segment that are relevant for the science data users are described.

## 1 Introduction

EarthCARE will provide global profiles of clouds, aerosols and precipitation along with co-located radiative flux measurements. Atmospheric microphysical properties and associated radiative fluxes will be used to evaluate the representation of
aerosols, clouds and precipitation in weather forecast and climate models and help improve their parameterisation schemes.

The EarthCARE satellite embarks four scientific instruments, an atmospheric lidar, a Doppler cloud radar, a multi-spectral imager and a broad-band radiometer. The instruments have been characterised, calibrated, tested and integrated on the spacecraft. The lidar operates at UV wavelength and is equipped with a high-spectral resolution receiver and depolarisation measurement channel. It will measure vertical profiles of aerosols and thin clouds and allow for classification of aerosol types.
The highly sensitive 94 GHz (W-band) cloud radar is equipped with Doppler measurement capability and will provide measurements of clouds and precipitation with a sensitivity partly overlapping with the lidar, but penetrating through or deep into





the cloud, far beyond where the lidar signal attenuates. Furthermore, it will measure the vertical velocities of cloud particles. The imager has four solar and three thermal channels and will provide across-track swath information on clouds and aerosols. It will also be used to construct retrieved 3D cloud/aerosol/precipitation scenes for radiative transfer calculations. The broad-
band radiometer measures solar and thermal radiances in three fixed viewing directions along the flight track, so that accurate top-of-atmosphere flux estimates can be derived. EarthCARE will fly in a sun-synchronic low-Earth orbit at a relatively low altitude in order to maximise the performance of the active instruments.

The EarthCARE retrieved geophysical data products will include, for example, target classification, vertical profile of microphysical properties of ice, mixed and liquid clouds, particle fall speed, precipitation parameters and aerosol type. By applying
1D and 3D radiative transfer models to those retrieved data, heating rates and radiative fluxes will be calculated. These calculations will be evaluated against the flux estimates derived from EarthCARE broad-band radiometer measurements. The scientific data products have been defined and their retrieval algorithms have been developed. The performance of the retrieval algorithms has been evaluated. Other papers in this Special Issue will describe the details of these data products and their retrieval algorithms.

The EarthCARE Payload Data Ground Segment (PDGS) produces data products up to the geophysical retrievals collocated at the satellite instruments' fields-of-view (Level 2 data products). (Higher processing levels and applications, such as data product exploitation for climate/weather model improvements, are out of the scope of this paper.)

This paper presents the EarthCARE mission background, the satellite with its scientific payload and the Ground Segment. The paper is intended as background information for the primary objective of this Special Issue, namely, the description of the
theoretical bases and validation of the various geophysical retrieval algorithms and data products developed in Europe, Japan and Canada for EarthCARE. Eisinger et al. (2022) describes the overall data processing chain and provides an overview of the data products.

Section 2 addresses the overarching scientific needs that justify the mission. The required observations are discussed in Section 3. A brief system overview is given in Section 4. The satellite instruments are described in Section 5. The orbit is
described in Section 6. The mission phases are described in Section 7. The mission ground segment, including a summary of the geophysical data products, is presented in Section 8.

## 2   Mission Science Objectives

EarthCARE will acquire key observables needed to better understand the role of cloud, aerosol and radiation on climate. The mission was selected in 2000 for pre-feasibility study, in 2001 for Phase A study, and in 2004 for implementation. Understand-
ing of the role of cloud radiative feedback and the direct and indirect radiative feedback of aerosols, and their representation in weather in climate models, has significantly improved since that time. Yet, high-resolution and vertically resolved observations of clouds and aerosols, co-located with measurements of reflected solar and emitted thermal radiation from Earth, remain critically important parameters to further the understanding of cloud and aerosol radiative impact and their feedback mechanisms in a warming climate.





The observations of the EarthCARE mission will address these science questions directly. The mission will measure (1) vertically resolved profiles of clouds, aerosols and precipitation at the satellite nadir track, (2) 3D cloud-aerosol scenes extending across-track and (3) co-located solar and thermal broad-band radiances and fluxes.

    Climate feedbacks from clouds are significant and by far the most inconsistent across climate models. The IPCC Sixth Assessment Report (IPCC 2021, in press) assumes a positive net cloud feedback in response to global warming with high con-

fidence. Major advances in the understanding of cloud processes, compared to the Fifth Assessment Report, have decreased the uncertainty range in the cloud feedback by about 50% and assume a net cloud feedback of +0.42 [-0.10 to 0.94] $Wm^{-2}K^{-1}$. Still, clouds remain the largest contribution to overall uncertainty in climate feedbacks. The IPCC report also addresses aerosol-cloud interaction and points out that "CMIP6 models generally represent more processes that drive aerosol–cloud interactions than the previous generation of climate models, but there is only medium confidence that those enhancements improve their

fitness-for-purpose of simulating radiative forcing of aerosol–cloud interactions." Finally – related to the objectives of Earth-CARE – the report acknowledges that CMIP6 models still have deficiencies in simulating precipitation patterns, particularly in the tropical ocean.

    The largest uncertainty of GCMs in the prediction of climate sensitivity to $CO_2$ increase is related to cloud feedback, due to the diversity of cloud formation processes and the feedback mechanisms depending on cloud macrophysical and microphysical

properties such as type, thickness, phase, location and height (Sherwood et al., 2020). In particular, height and vertical extent of cloud over the globe, which are essential for cloud feedback evaluation, cannot be satisfactorily determined by passive space sensors, let alone ground-based and airborne observations, but require the use of space-borne active, range-resolving, lidar and radar systems. Regarding detection of cloud feedback to a warming climate, Takahashi et al. (2019) pointed out that, using W-band radar, long data records extending into the 2030s are needed to reliably detect trends in cloud heights. Vaillant de Guélis

et al. (2018) stressed that traditional satellite cloud records are poorly constraining cloud feedback due to the lack of vertical cloud profile information and, therefore, cloud lidars are required to reduce climate sensitivity uncertainties. Similarly, Chepfer et al. (2014) calls for a long-term lidar record in order to detect changes in clouds. Substantial advances in the understanding of interactions of cloud and climate were achieved with CloudSat and CALIPSO, but will have to be continued in order to allow trend detection (Winker et al., 2017).

Illingworth et al. (2015) provided an overview of the EarthCARE mission and its scientific goals. As addressed in the paper, the objective of the mission is to provide observations of global cloud and aerosol profiles together with radiation observations, in order to provide data for the improvement of clouds and aerosol parameterisation in climate and weather models. Required observations include global vertical profiles of natural and anthropogenic aerosol, vertical profiles of ice and liquid cloud, precipitation and derived vertical heating rates. EarthCARE's observed top-of-atmosphere radiances and fluxes

will both contribute to Earth radiation budget studies and, more importantly, be used in verification of the calculated heating rate and flux profiles calculated from retrieved cloud-aerosol scenes. Beyond these original EarthCARE mission objectives related to climate and numerical weather prediction (NWP) model improvements, it is now envisaged by ECMWF to assimilate EarthCARE lidar and radar observations operationally into the Integrated Forecasting System in anticipation of a positive impact on the forecast accuracy (Janisková and Fielding, 2020).



EarthCARE will both extend the record of the former space-borne lidar and radar observations of CloudSat and CALIPSO and provide increased performance of such observations, by using a lidar with high-spectral resolution and polarisation channel and an accurately co-located W-band radar with Doppler capability – which will be the first in space – and an increased sensitivity of about 5dB compared to CloudSat. With the help of a push-broom imager, 3D cloud-aerosol scenes will be retrieved, and heating rate and radiative flux profiles will be calculated and compared to co-located top-of-atmosphere fluxes derived from the broad-band radiometer measurements.

EarthCARE also builds on the success of the CloudSat and CALIPSO missions of the A-Train (Stephens et al., 2018). Launched in 2006, both satellites have exceeded their design life time by far and delivered unprecedented measurements of cloud and aerosol profiles and precipitation. In comparison to the pre-CloudSat/CALIPSO era, when only passive space borne cloud observations were available, the understanding of cloud net forcing has been significantly improved through provision of true measured cloud heights and optical thicknesses derived from measured ice and water paths and the identification of multi-layered cloud systems. Stephens et al. (2018) provides an overview of how CloudSat/CALIPSO measurements provide a new understanding of clouds, their processes and radiative effects. Winker et al. (2013) describes how CALIPSO observations were used to characterise the global 3D distributions of aerosol and their seasonal and inter-annual variations, providing unprecedented data sets for comparisons to aerosol models.

EarthCARE will extend and advance these observations with a radar with Doppler capability to give information on convective motions as well as ice and rain fall speeds, leading to improved drizzle, rainfall, and snowfall rates. The additional sensitivity compared with CloudSat will enable it to better detect thin ice clouds and much more low-level stratus and stratocumulus. The lidar will operate in the UV and is equipped with a high-spectral resolution receiver to discriminate particulate and molecular backscatter, thus providing direct measurements of the extinction profile of clouds and aerosol plus a polarisation channel for advanced classification of aerosol type and ice particle characteristics.

Earth Radiation Budget missions are also relevant heritage for EarthCARE, so that measured cloud and aerosol profiles can be utilised in the context of co-located radiative flux measurements. Namely, CERES (Wielicki et al., 1996) observations have been used to evaluate radiative transfer properties derived from CloudSat and CALIPSO cloud and aerosol retrievals (Kato et al., 2011). EarthCARE will fly its own broad-band radiometer to derive top-of-atmosphere radiative fluxes (Velázquez-Blázquez et al., 2022b), allowing for similar studies.

## 3  Observational Requirements

The EarthCARE Mission Requirements Document (Wehr, 2006, MRD) was written after the completion of the feasibility study (Mission Phase A). The MRD defines the scientific mission objectives and observational requirements as a reference for the system (including satellite, instruments, ground segment, data products) design and implementation.

The MRD prescribes that EarthCARE shall meet these mission objectives by simultaneously measuring the vertical structure and horizontal distribution of cloud and aerosol fields together with outgoing radiation over all climate zones. It shall measure properties of aerosol layers (occurrences, profile of extinction, boundary layer height, aerosol type), clouds (3D cloud



boundaries, occurance of ice/liquid/super-cooled layers, vertical profiles of ice and liquid water content along with effective ice particles and shape or effective droplet size, respectively, vertical velocities (cloud convective motion and ice sedimentation),

drizzle and rainfall rates, co-located with top-of-atmosphere solar and thermal fluxes.

The key accuracy requirement provided in the MRD stipulates that a retrieved (cloud-aerosol-precipitation) scene with a footprint size of 10 km × 10 km shall be sufficiently accurate, so that its atmospheric vertical profile of short-wave (solar) and long-wave (thermal) flux can be reconstructed with an accuracy of $10\,\mathrm{Wm^{-2}}$ at the top of atmosphere.

Therefore, the geophysical data products retrieved by EarthCARE must provide, firstly, a comprehensive set of cloud and

aerosol vertical properties for each observed scene. Secondly, these scenes must be extended in the across-track dimension, in order to establish 3D scenes sufficiently wide for the application of – depending on the scene complexity – 1D or 3D Monte-Carlo radiative transfer calculations providing vertical profiles of heating rates and (broad-band) solar and thermal fluxes, co-located with the retrieved cloud/precipitation and aerosol profiles. The so-derived fluxes (and radiances) at the top of atmosphere are, thirdly, compared to the measurements of the EarthCARE broad-band radiometer for verifica-

tion of the cloud-aerosol retrievals and consistency of the radiative transfer calculations. The required spatial resolution for cloud/precipitation/aerosol profiles along-track is typically 1 to several kilometres and the vertical resolution 100 to a few hundred metres, depending on the target. The radiative properties are evaluated for scenes of $100\,\mathrm{km^2}$ size.

Table 1 lists the MRD observational requirements for clouds and aerosols. The requirements are derived from the $10\,\mathrm{Wm^{-2}}$ requirement, with the exception of the vertical velocity requirement. For this, the MRD requires the measurement of vertical

motion within clouds, in particular for characterising convection, estimating sedimentation velocity of ice particles in cirrus, quantifying drizzle fluxes in stratocumulus and estimating heavier rain rates. The MRD requires retrieved cloud products on a horizontal grid of 10 km, while individual measurements and retrievals should be made on a 1-km grid.

Global sampling is required at a fixed orbit local time in order to avoid aliasing of diurnal and seasonal variations. The choice of the orbit equatorial crossing time is a compromise depending on both instrument performaces and diurnal variations

of atmospheric properties and radiation fields. Afternoon orbits are generally preferred, as convection (over land) is mainly initiated in the early afternoon. Furthermore, near-noon retrievals of top-of-atmosphere broad-band short-wave fluxes are most representative of the diurnal average and relative errors of short-wave radiance measurements and flux retrievals are smallest, when the short-wave signal is largest. In order to minimise the effect of sunglint, the orbit must not be too close to noon. The MRD requires an equator crossing time between 13:15 and 14:00 hours.

In order to fully exploit the synergistic value of the observations of the EarthCARE instruments, their footprints must be co-located. The MRD requires that three instruments used for cloud, precipitation and aerosol retrievals are co-located within 350 m (goal 200 m) and with the radiation measurements within 1 km.

The data latency for Level 1 data products (sub-section 8.2) has to be compatible with monitoring and data assimilation into NWP models. When the MRD was written in 2006, advances in NWP modelling were anticipated that would allow

for the assimilation of EarthCARE data, which has been confirmed in the mean time (Janisková and Fielding, 2020). For climatological studies and evaluation of climate and NWP models, delayed data latencies of several days to a month might be





| Property (Cloud) | Detectability Threshold | Accuracy |
|---|---|---|
| Ice cloud top/base | N/A | 300 m |
| Ice cloud extinction coefficient | 0.05 km$^{-1}$ | 15% |
| Ice water content | 0.001 gm$^{-3}$ | $\pm$ 30 % |
| Ice crystal effective size | N/A | $\pm$ 30 % |
| Water cloud top/base | N/A | 300 m |
| Water cloud extinction coefficient | 0.05 km$^{-1}$ | 15 % |
| Liquid water content | 0.1 gm$^{-3}$ | $\pm$ 15 to 20 % |
| Water droplet effective radius | N/A | $\pm$ 1 to 2 $\mu$m |
| Fractional cloud cover | 5 % | 5 % |
| Vertical velocity within clouds | N/A | $\pm$ 0.2 to 1 ms$^{-1}$ |
| Property (Aerosol) | Detectability Threshold | Accuracy |
| Boundary layer optical depth | 0.05 | 10 to 15 % |
| Top/base and profile | N/A | 500 m |

**Table 1.** Retrieval accuracy requirements according to the MRD. Detectability is defined as the value measured with 100% RMS error.

adequate, however, a fast data availability for Level 1 and Level 2 data would still be very important for fast detection of any possible data degradation, so that corrective actions can be taken.

The EarthCARE mission has been developed in full compliance with its observational requirements. The satellite and in-
strument performances have been verified and confirmed. A sophisticated data processing system has been developed and the comprehensive data products and their retrieval algorithms – presented in this Special Issue of this journal – is fully compliant with the EarthCARE observational requirements.

## 4   EarthCARE System Overview

The EarthCARE satellite is a three-axis stabilised, custom build carbon-fiber reinforced polymer (CFRP) platform, optimised
for minimum mass and high stability. The streamlined shape of the spacecraft, with its trailing solar panel, minimises its cross section in order to reduce residual atmospheric drag at its relatively low flight altitude of 393 km (nominal mean spherical altitude). The low altitude has been selected in order to optimise the performance of the two active instruments, the lidar and the radar. The orbit is sun-synchronous, with a descending node crossing time of 14:00 hours, an inclination of 97° and revisit time of 25 days (389 orbits). The deployable solar array delivers a power of 1710 W (average at end of life) for a
nominal average power requirement of 1670 W and use of Lithium-Ion batteries with a capacity of 324 Ah. The total mass of the satellite is 2350 kg including 313 kg of hydrazine on-board propellant, sufficient for three-years operation plus one year reserve.



The satellite embarks four scientific instruments. The ATmospheric LIDar (ATLID) is a cloud-aerosol high-spectral res-
olution lidar (HSRL) operating at 355 nm wavelength. It is equipped with a receiver separating molecular from particulate
backscatter and depolarised total backscatter. It will provide measurements of the vertical profiles of aerosol and (thin) clouds.
The Cloud Profiling Radar (CPR) is a 94 GHz (W-band) cloud radar with Doppler capability, that will provide cloud profiles,
rain estimates and particle vertical velocity. The Multi-Spectral Imager (MSI), with four solar and three thermal channels,
provides cloud and aerosol observations extended in the across-track direction. MSI is essential for creating 3D cloud-aerosol
scenes out of the lidar and radar derived vertical profiles, in order to calculate radiative properties of these scenes, including
heating rates and top-of-atmosphere radiances and fluxes, which will be compared to those measured by the fourth EarthCARE
instrument, the Broad-Band Radiometer (BBR). The Satellite, including the instruments ATLID, MSI and BBR is developed
by ESA, with Airbus Space and Defence GmbH, Germany, as Prime Contractor, while the CPR is developed by JAXA and the
National Institute of Information and Communications Technology (NICT) in Japan.

The satellite command and control uses S-band up-/downlinks at the ground station in Kiruna (Sweden). An X-band system
is used for science data communication using both Kiruna and Inuvik (Canada) ground stations. The science data packages
are sent to the Payload Data Ground Segment (PDGS) at ESA/European Space Research Institute (ESRIN), Frascati, for the
production of Level 0 data. Subsequent Level 1 and Level 2 operational processing is performed at JAXA for CPR Level 1 and
JAXA Level 2 data products, and at ESA for ATLID, MSI, BBR Level 1 and ESA Level 2 data products. Both ESA and JAXA
Level 1 and Level 2 data products will be made available to the global user community.

The sophisticated data processing system is described by Eisinger et al. (2022). The geophysical data products and their
retrieval algorithms are described in the papers of this Special Issue.

## 5   Satellite Instruments

The satellite science payload consists of four instruments: The ATmospheric LIDar (ATLID), the Cloud Profiling Radar (CPR),
the Multi-Spectral Imager (MSI) and the Broad-Band Radiometer (BBR). Figure 1 shows an artist impression of the satellite
and indicates the location of the four instruments. Figure 2 shows a schematic sketch of the viewing geometry of the instru-
ments. The satellite instruments have been previously described, in particular by Wallace et al. (2014, 2016); Heliere et al.
(2017). The CPR has been described previously by Nakatsuka et al. (2012). However, meanwhile, some details of the instru-
ments have changed and their calibration and performance verifications have been completed.

### 5.1   The ATmospheric LIDar - ATLID

ATLID is a high-spectral-resolution (HSR) atmospheric backscatter Light Detection and Ranging (LIDAR) instrument (HSRL)
for the detection of cloud boundaries and profiling of optically thin clouds and aerosols. The instrument has been developed by
Airbus Defence and Space (France).

ATLID is used for the retrieval of a lidar feature mask (Nishizawa et al., 2022; van Zadelhoff et al., 2023) and, subsequently,
cloud and aerosol profile retrievals (Donovan et al., 2022b; Nishizawa et al., 2022; Okamoto et al., 2022; Wandinger et al.,





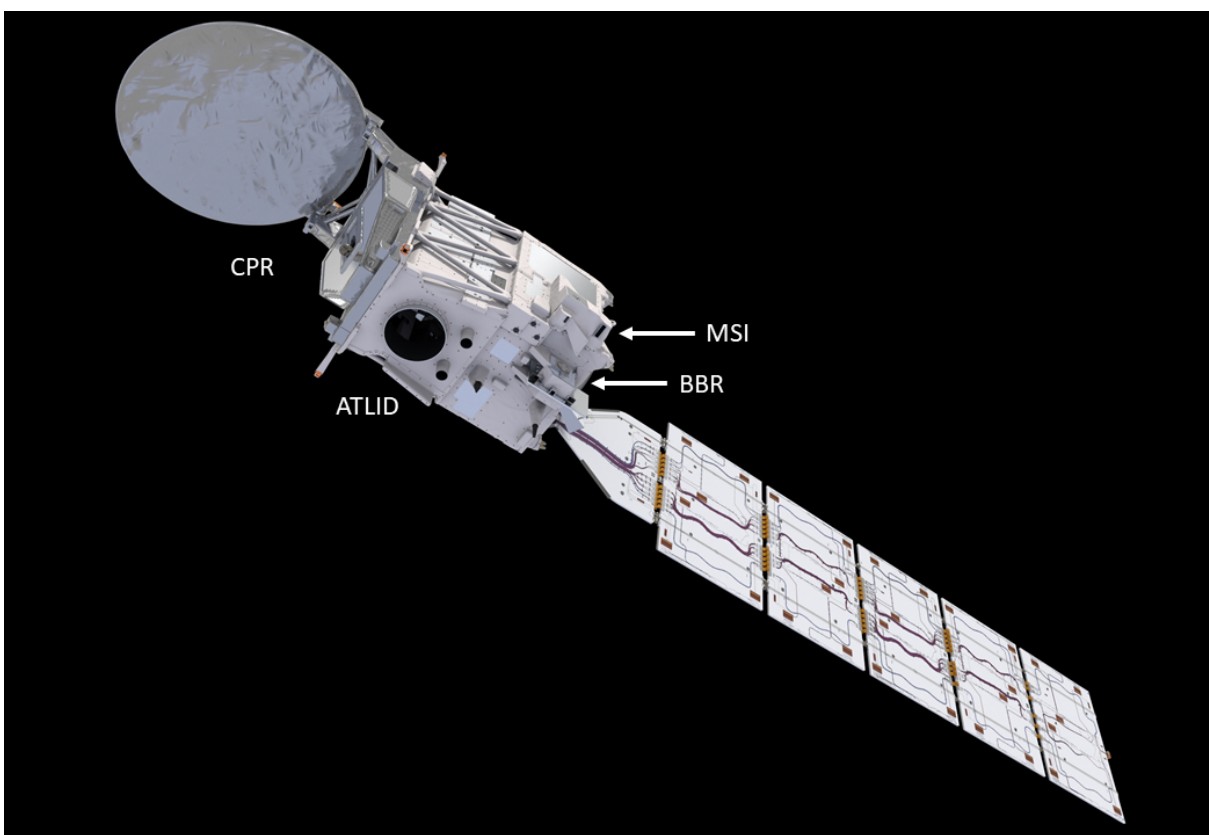

**Figure 1.** Artist impression of the EarthCARE satellite with the location of the four science instruments. (ESA and ATG Medialab, The Netherlands)

2022b) and, in synergy with MSI data, to derive cloud top height/column properties (Haarig et al., 2022) and aerosol properties (Kudo et al., 2016). ATLID is furthermore used in synergy with CPR and MSI to derive synergistic target classification and atmospheric profile products (Irbah et al., 2022; Mason et al., 2022a; Okamoto et al., 2022).

ATLID is designed as a self-standing instrument, reducing the mechanical coupling of instrument/platform interfaces and allowing for better flexibility in the satellite integration sequence. The transceiver architecture selected for this instrument is bistatic, meaning that the two redundant emission chains are separated from the receiver chain. Consequently, there is a single optical element exposed to vacuum that is illuminated by the laser. This approach was selected in order to minimising the risk of Laser Induced Contamination (LIC) of optical surfaces. Figure 3 shows a schematic sketch of the bistatic design, including the two laser sources.

In order to ensure the longevity of the mission (minimum 2.5 years of operations after commissioning), there are two completely independent, cold redundant emission chains. Each emission chain is manufactured by LEONARDO (Pomezia, Italy) and henceforth is referred to as a Transmitter Assembly (TxA). Each TxA is comprised of a Reference Laser Head (RLH),





**Figure 2.** Illustration of viewing geometry (not proportional). The CPR is pointing exactly nadir (indicated in green). The radar footprint is approximately 700 m (3 dB antenna beam). ATLID is pointed 3° off-nadir backwards along track to minimise specular reflection from ice crystals. Its telescope footprint is <30 m. The MSI swath is 150 km wide, tilted away from the sun glint affected side, so that it extends 35 km to one side of nadir and 115 km to the other. The BBR has three fixed telescopes, forward, nadir and back-wards pointing. The scene size is configurable. The nominal size is 10 km × 10 km, but a size of 5 km wide and 21 km long will be used for radiative transfer calculation and closure assessment (Cole et al., 2022). The BBR fore- and aft-views are pointing 50° forward and backwards, respectively, leading to a zenith angle on ground of 54-55°.

developed by TESAT (Backnang, Germany), seeding via optical fiber a Power Laser Head (PLH), plus the associated Transmitter Laser Electronics (TLE). A major evolution with respect to the ALADIN (Cosentino et al., 2012) laser of the Aeolus mission (Straume, A.G. et al., 2020), the ATLID PLH is a sealed and pressurised enclosure at 1.2 Atm dry air (pressurisation verified for over 10 years of space operations). This approach, in addition to careful selection and screening of included materials as well as extensive bake out of all optical assemblies, ensures a mitigation of LIC (Wernham et al., 2010). A two year dedicated test campaign (accelerated test extrapolated to lifetime duration via contaminant heating and higher pulse repetition



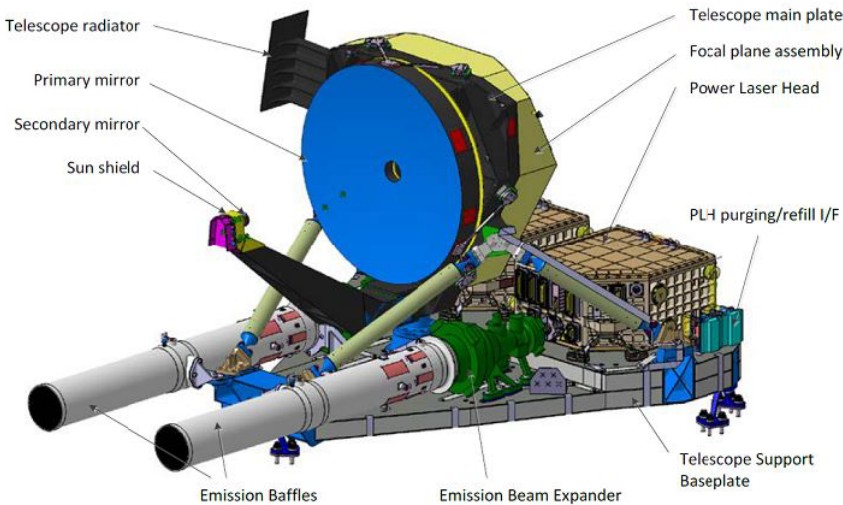

**Figure 3.** Illustration of ATLID bistatic architecture. This graphic representation shows the two fully redundant transmitting chains including their emission telescopes. The long emission baffles at the exit of the two laser heads minimise risk of LIC for the exit windows. (Courtesy of Airbus DS, France.).

frequency) demonstrated the advantages of this approach. The PLH design has a Master Oscillator Pump Amplifier (MOPA) laser architecture. The seeded Master Oscillator (MO) is an end pumped Nd:Yag rod, pumped at 808nm by collimated high power laser diode stacks. The MO is equipped with 1+1 cold redundant diodes stacks, to compensate diode ageing losses, and is isostatically mounted on a thick, dedicated optical plate that is totally decoupled from all thermal sources. The MO is significantly derated (operates at one third of its peak power), in order to provide margin for possible energy recovery actions, as well as to extend the lifetime of its optical coatings. Special care was taken to ensure the entire laser is operating at low optical fluence, as coating longevity is significantly affected by high fluence. The pulse generated by the MO is amplified by a double pass Pump Amplifier before passing to a frequency tripling conversion crystal set to a wavelength of 355 nm. Another change to the design, with respect to the ALADIN laser, is the UV section. The beam coming out of the Third Harmonic Generator is expanded immediately, to ensure very low fluence in the UV section, and the number of optical elements in UV are kept to a minimum (and partly enclosed with an open cover to minimise the field-of-view of possible contaminants). Finally, the laser unit is equipped with 5 photodetectors that allow precise knowledge of the emitted power (measurement accuracy 2%) to permit control of the emitted frequency of the laser. The output from this laser is a linearly polarised laser pulses, of smaller than 35 ns pulse length, at a pulse repetition rate of 51 Hz and with a 31 to 35 mJ pulse energy. The output beam size is approximately 8x9 mm$^2$ at laser exit, with a divergence of 36 $\mu$rad after ×6.7 beam expander. It points 3° backwards of nadir in order to minimise specular reflection on ice clouds. Figure 5 plots the UV energy during the PLH FS Life Time Test performed by LEONARDO.





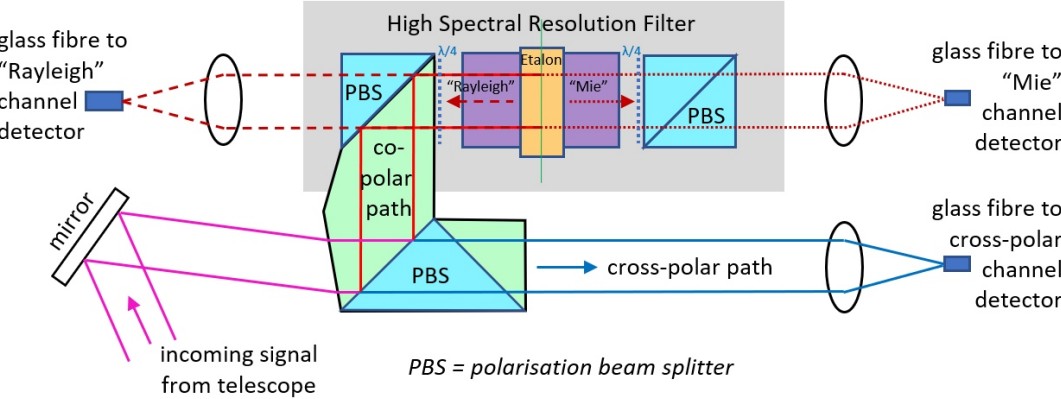

**Figure 4.** ATLID receiver optical path schematic. Returned signal collected from the telescope is passing through a Polarisation Beam Splitter and the total cross polar signal is separated to the cross polar path (via fiber, to end up in the cross polar channel detector). The remaining co-polar signal enters the HSR filter, that separates the Mie from the Rayleigh in individual polar channels.

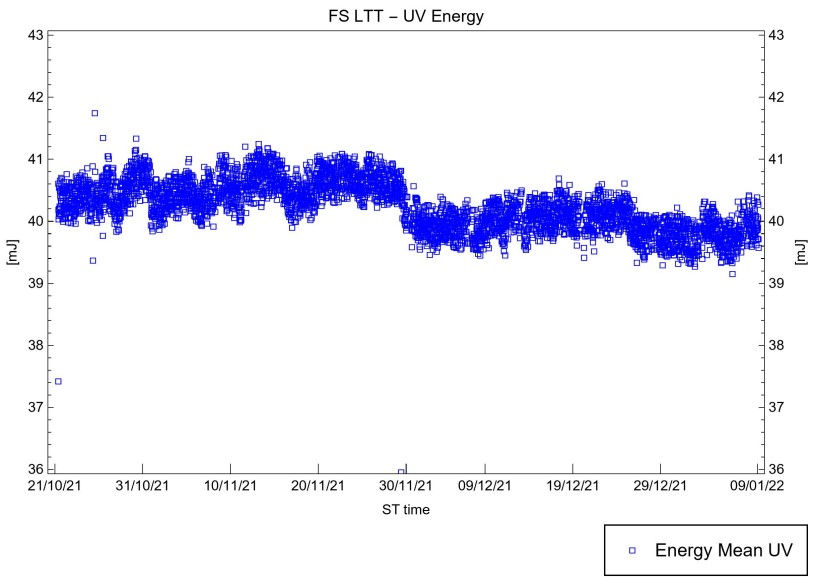

**Figure 5.** This test was performed on the spare PLH and accumulated over 6 months of operation. The energy loss was less than 0.3 mJ, in line with diode stack degradation (without any compensation). The variation is due to significant lab temperature variations that doesn't impact instrument performance, as the energy is constantly and accurately monitored by internal telemetry. Upon completion of the test, the PLH was refurbished by replacing the Pump Unit as well as some UV optics to mitigate possible ageing effects. (Courtesy of LEONARDO, Italy.)

Figure 4 shows schematically the optical path of the ATLID receiver. Light generated by the TxA and backscattered in the atmosphere is collected by an afocal Cassegrain telescope with a diameter of 62 cm. The ATLID receiver separately





measures the co-polar and cross-polar back-scattered signals. The co-polar signal is further separated, using a High spectral Resolution Etalon (HSRE) developed by Thales Alenia Space (Switzerland), into a spectrally broadened molecular backscatter ("Rayleigh") signal channel and a particulate backscatter channel (usually physically inaccurately referred to as "Mie"), ac-

cording to the principle of a high-spectral resolution lidar (HSRL). Therefore, ATLID has three receiver channels, referred to as the *Cross-Polar Channel*, and the *Rayleigh* and *Mie Channels* (which are both co-polar).

Figures 6 and 7 show the Fabry Perot Etalon design and its Mie channel HSR module spectral proportion. The separated signals from the 3 channels are coupled into fibers and routed to 3 identical and separate Memory Charge - Coupled Devices (MCCDs). The MCCD, developed by Teledyne e2V (UK), is able to measure single photon events to meet the worst case

radiometric performance requirements. The selected design provides high response, together with an extremely low noise thanks to on-chip storage of the echo samples, which allows delayed read-out at very low pixel frequency (typically below 50 kHz). Combined with an innovative read-out stage and sampling technique, the detection chain provides an extremely low read-out noise (< 2e- rms per sample). With respect to the hot pixel issues observed in AEOLUS mission (Weiler et al., 2021), ATLID pixel level CCD behaviour has been characterised under flight-like operational conditions and the dark current values

do not present the same level of defect, with no hot pixels observed; this is due to a different CCD control sequence, which reduces the time that the charge resides in the device by a factor of 10. Each detector achieves a vertical resolution of 103 m in the vertical range from the surface up to 20 km altitude and of about 500 m in the vertical range of 20 km up to 40 km. The along-track sampling distance is 140 m, however, in order to increase the signal-to-noise performance, it is envisaged to integrate two consecutive lidar pixels, leading to an effective along-track spatial resolution of about 280 m.

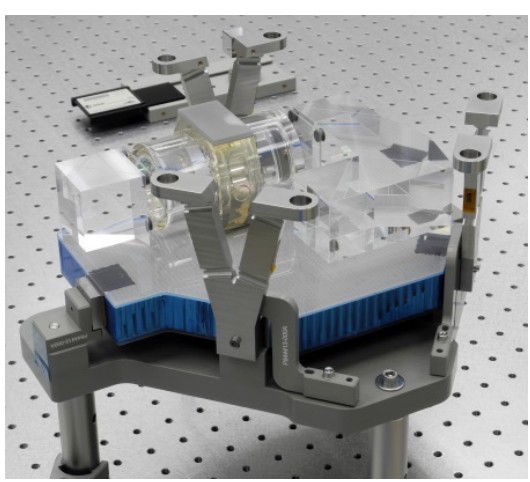

**Figure 6.** Fabry Perot Etalon (FPE). (Courtesy of Thales Alenia Space, Switzerland.)

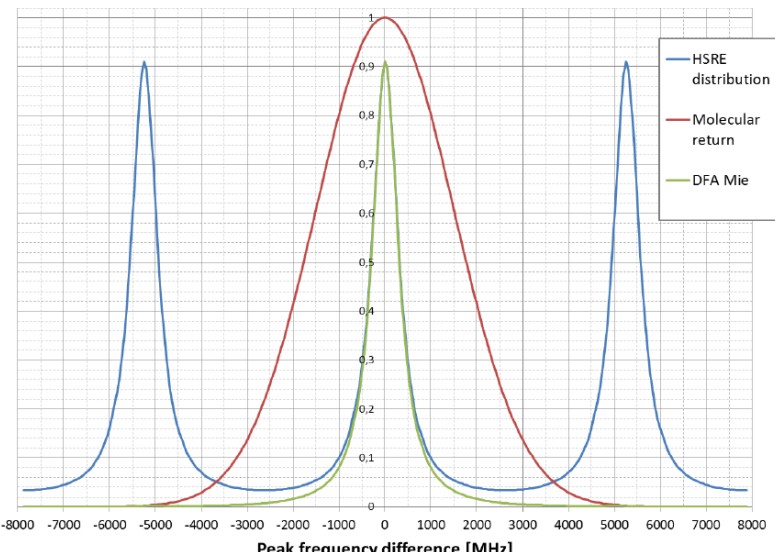

**Figure 7.** Mie channel HSR module spectral proportion (atmosphere at 248 K). HSRE = High Spectral Resolution Etalon. (Courtesy of Airbus DS, France.)



In order to align the emission beam to the telescope FoV the instrument implements a co-alignment control loop. Each PLH contains a Beam Steering Mechanism (BSM) that is controlled by the Co-Alignment Sensor (CAS) in the receiver chain (a beam splitter divides off approximately 7% of the incoming signal towards the co-alignment detector prior to the polarisation separation). This active alignment approach allows to correct for depressurisation, structural as well as thermo-elastic deformations that could be expected after launch or during orbit. The MCCD sensor used in the CAS is identical to the 3 other ATLID channels but operates uncooled, so th performace is degraded.

The ATLID instrument at launch is fully compliant with the performance requirements for the current orbit selection. The instrument was designed with significant performance margins in order to remain compliant even at worse case end of mission, including the design case high orbit of up to 425 km. Table 2 summarises the most important performance requirements, as well as the measured beginning of life and extrapolated worse case at end of life conditions at the nominal operational orbit. The variations of the two lasers are caused by the accuracy of measuring the emitted power. In this aspect ATLID project benefited significantly from the lessons learned from ALADIN, with improved monitoring setup and the optics after the UV monitoring photodiode kept to the minimum. Figures 8 and 9 provide a more detailed plot of the channel retrieval accuracies versus latitude during summer as well winter conditions.

In order to achieve the above mentioned performances, a number of instrument calibrations take place systematically on orbit. Some of them are adapted with every pulse, while other calibrations require the instrument to switch into a specific mode (Dark Current Calibration requires the laser to be switched off). The accuracy of each derived input signal is not a product of one calibration, but a result of several in flight calibrations as well as an a-priory good knowledge of the instrument characteristics and capabilities. The calibration requirements for absolute calibration accuracies are derived from the periodic measurement of total noise in darkness, the pulse to pulse background subtraction accuracy, the systematic calculation of lidar constants on both channels, as well as the spectral cross-talk knowledge accuracy. Furthermore, each type of calibration performed requires a specific observation scene as well as certain conditions to be met (e.g. preference of night measurement, avoidance of South Atlantic Anomaly, specific cloud formations, avoidance of ocean overflights etc). Figure 3 summarises the most critical calibration requirements as well as the spectral and polarisation cross talk channel knowledge.

The pre-launch testing and calibration of ATLID has been described by do Carmo et al. (2021).

The ATLID calibration algorithms (Level 1 processing) and Level 1b data products are described by Eisinger et al., 2022. The Level 1 data product consists of geolocated attenuated particulate ("Mie") backscatter, attenuated molecular ("Rayleigh") backscatter and attenuated cross-polar backscatter.

## 5.2 The Cloud Profiling Radar - CPR

EarthCARE CPR is a 94 GHz pulse radar which measures altitude and Doppler velocity from the received echo. It is the world's first space-borne cloud radar that estimates vertical profiles of the cloud and its vertical motion. Its minimum sensitivity is better than -35 dBZ with the Doppler velocity measurement accuracy of better than 1.3 m/s for specification and also 1.0 m/s for target performance. To achieve those performances, the CPR is equipped with a large reflector with a diameter of 2.5 m. The instrument has three observation modes: Low (16 km), Middle (18 km), High (20 km). The instrument has been





| Performance Requirements | Typical BOL PFM | Typical BOL FM | WC at EOL PFM | WC at EOL FM | Requirement value |
|---|---|---|---|---|---|
| Mie Co polar Accuracy *(cirrus @10 km altitude, dense cloud @4 km, horizontal av. 10 km)* | 33.3% | 30.1% | 46.5% | 47.1% | 48% |
| Rayleigh Accuracy *(above cirrus @10 km altitude, dense cloud @4 km, horizontal av. 10 km)* | 10.9% | 10.4% | 14.1% | 14.6% | 15% |
| Cross polar Accuracy *(cirrus @10 km altitude, dense cloud @4 km, horizontal av. 10 km)* | 15.0% | 15.3% | 16.9% | 18.5% | 45% |
| Rayleigh Accuracy in PBL *(horizontal av. 10 km)* | 5.7% | 5.4% | 7.4% | 7.5% | 10% |
| Radiometric Stability | Rayleigh 0.62% | Mie co 0.77% | Rayleigh 1.1% | Mie co 3% | <5% Mie co <2% Rayleigh |

**Table 2.** A summary of the instrument performance requirements for both transmitters (PFM and FM) for typical beginning of life (BOL) operations as well as worst case (WC) scenario at end of life (EOL). The delta in the performances between the two transmitters is driven by the Energy measurement accuracy. For Mie Co-polar accuracy $\beta = 8 * E10^{-7} sr^{-1} m^{-1}$, while for Rayleigh Accuracy at PBL $8 * E10^{-7} sr^{-1} m^{-1} \leq \beta \leq 7 * E10^{-6} sr^{-1} m^{-1}$.

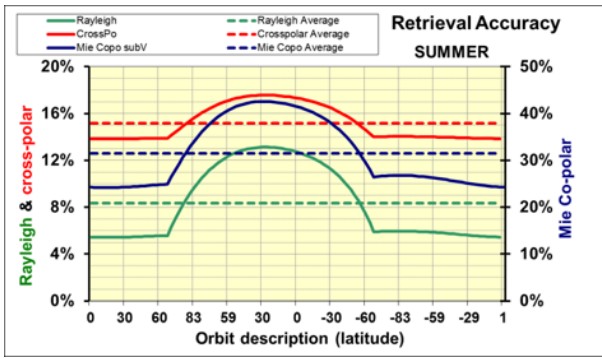

**Figure 8.** ATLID channels retrieval accuracy versus latitude during Summer. (Courtesy of Airbus DS, France.)

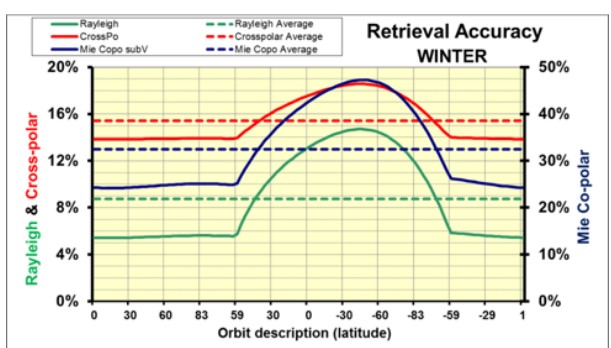

**Figure 9.** ATLID channels retrieval accuracy versus latitude during Winter. (Courtesy of Airbus DS, France.)





| Calibration Requirements | Requirement | WC EOL |
|---|---|---|
| Mie Co-polar Absolute calibration accuracy | 10% | 11.1% |
| Mie Cross-polar Absolute calibration accuracy | 15% | 13.1% |
| Rayleigh Absolute calibration accuracy | 10% | 6.6% |
| *Presented figures during daylight. No non compliances during eclipse.* | | |
| Rayleigh calibration above 30km and background above 100km | | compliant |
| Spectral X-talk knowledge in the Rayleigh channel | 20% | 7.9% |
| Spectral X-talk knowledge in the Mie co-polar channel | 10% | 9.6% |
| Polarisation X-talk knowledge in the Mie co-polar channel | 0.01 | 0.012% |
| Spectral X-talk knowledge in the Mie cross-polar channel | 0.01 | 0.012% |
| Instrument shall transmit all ancillary data required for the on-ground data processing | | compliant |

**Table 3.** Instrument Calibration requirements, as well as Spectral and Polarisation cross-talk between the 3 channels. (WC EOL = worst case at end of life.)

jointly developed by Japan Aerospace Exploration Agency (JAXA) and National Institute of Information and Communications
Technology (NICT).

The CPR Level 1 product contains radar reflectivity and Doppler velocity detected from cloud and precipitation particles. Those products are used to derive cloud mask and cloud and precipitation property (Okamoto et al., 2022; Kollias et al., 2022; Mroz et al., 2023). In synergistic use with ATLID and MSI, those data are also used to classify atmospheric targets (Irbah et al., 2022) and to derive those profiles that are important for radiative transfer calculation (Mason et al., 2022a).

The external view of the EarthCARE CPR and major components layout inside the platform are shown in Fig. 10. Also a simple block diagram of the CPR is shown in Fig. 11. The CPR consists of TRS (Transmitter Receiver Subsystem) and SPU (Signal Processor Unit), QOF (Quasi Optical Feeder) with sub reflector, and MREF (Main Reflector). The TRS mainly consists of LPT (Low power transmitter), HPT (High power transmitter) and RCV (receiver): The 94 GHz pulse signal is generated by the LPT, the signal is amplified by the HPT to more than 1.5kW using an EIK (Extended Interaction Klystron) and the signal
is transmitted towards the QOF. The RCV receives echoes from clouds and ground through the QOF and down converts to an IF frequency, which is 60 MHz, and the IF signal is transferred to the SPU (Signal Processor Unit). The TRS is a fully-redundant system, in particular, the HPT is linked with both nominal and redundant sides via a magic T which is a 3dB hybrid waveguide coupler used for power divider to both sides of the HPT. The leak signal from the QOF during the transmitting of the signal pulses is used as reference for the Doppler capability. The noise source (hot and ambient) within the RCV, which
can be switched from the antenna (observation) receiving channel, is used to calibrate the receiver system. The QOF has two functions of primary feeds to the MREF and separation of the Tx (transmitting) and Rx (receiving) signals. Input signal to the QOF, which is linearly polarized, is transformed to circular polarization by means of an FSP (Frequency Selective Polarizer) system in the QOF. A LHCP (left hand circular polarization) signal is transmitted from the CPR to the Earth, and a RHCP (right





hand circular polarization) signal is returned to the CPR. The diameter of the MREF aperture is 2.5 m. This size was derived
from the restriction of the fairing size of the assumed rocket launcher and the need to maximize the antenna gain and therefore
decrease the estimated error in Doppler velocity retrieval. To Manufacture a large antenna reflector of 2.5 m was challenging
for the frequency band and thus an accurate and light-weighted antenna was demanded for space borne millimeter-wave radars.

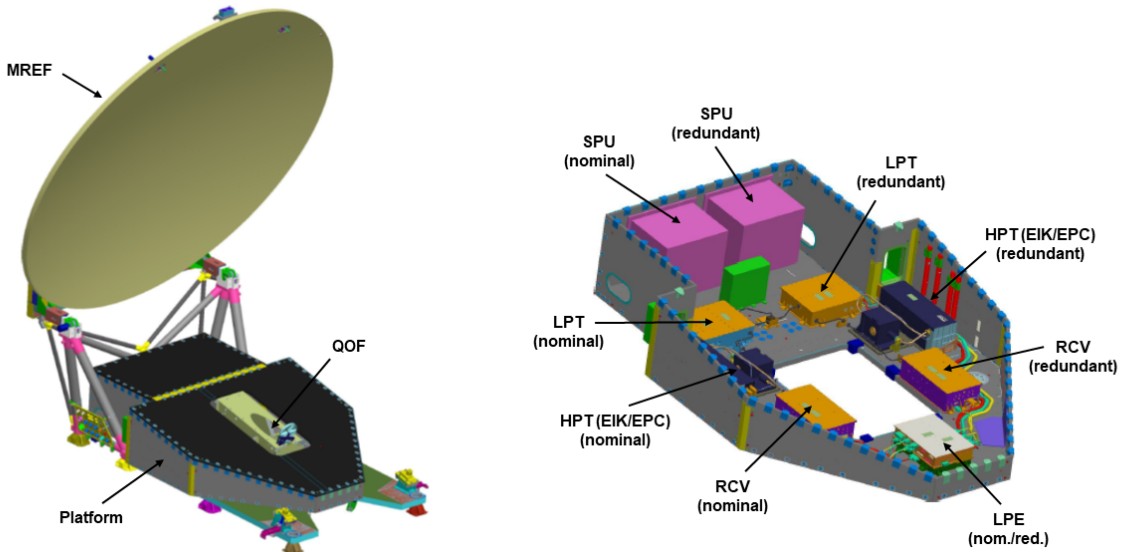

**Figure 10.** CPR external view with MREF deployed (left) and major components layout inside Platform with upper panel opened (right).
(Courtesy of NEC)

The frequency of radar signal is 94GHz which is same as the CloudSat CPR (Stephens et al., 2008), but the sensitivity
of the radar is much better due to its lower orbit and larger antenna aperture size. The minimum radar reflectivity of the
EarthCARE CPR is -35 dBZ defined at the top of atmosphere (20 km) under the condition of 10 km horizontal integration.
The observation range of the Doppler velocity is ±10 m/s, with an accuracy of at least 1.3 m/s for specification and also 1.0
m/s for target performance in the case ideal beam pointing error (bias and harmonic elements) removal achieved in orbit, for
cloud echoes of more than -19 dBZ when they are integrated over a 10 km horizontal distance. The transmit pulse width is
3.3 microseconds corresponding to the vertical resolution of 500m, which is also the same as the CloudSat CPR. However, the
received echo is over-sampled at 100 m, which is more precise than that of CloudSat (250 m). The vertical observation range
can be selected from three different range window settings of 20, 18, or 16 km considering on the satellite latitude, and the
lowest measurement altitude extends to -1 km (below the ground surface), allowing the utilization of surface backscatter. To
Minimize the range window would be needed to achieve a higher PRF (pulse repetition frequency) for more accurate Doppler
measurement. A variable PRF function has been implemented for altering the PRF with satellite altitude and latitude. The
instantaneous footprint size of the CPR is about 750 m at ground level under the assumption of a maximum geodetic altitude
of 450 km. Horizontally-averaged data along track is produced with the minimum pixel size of 500 m for a fine SNR and





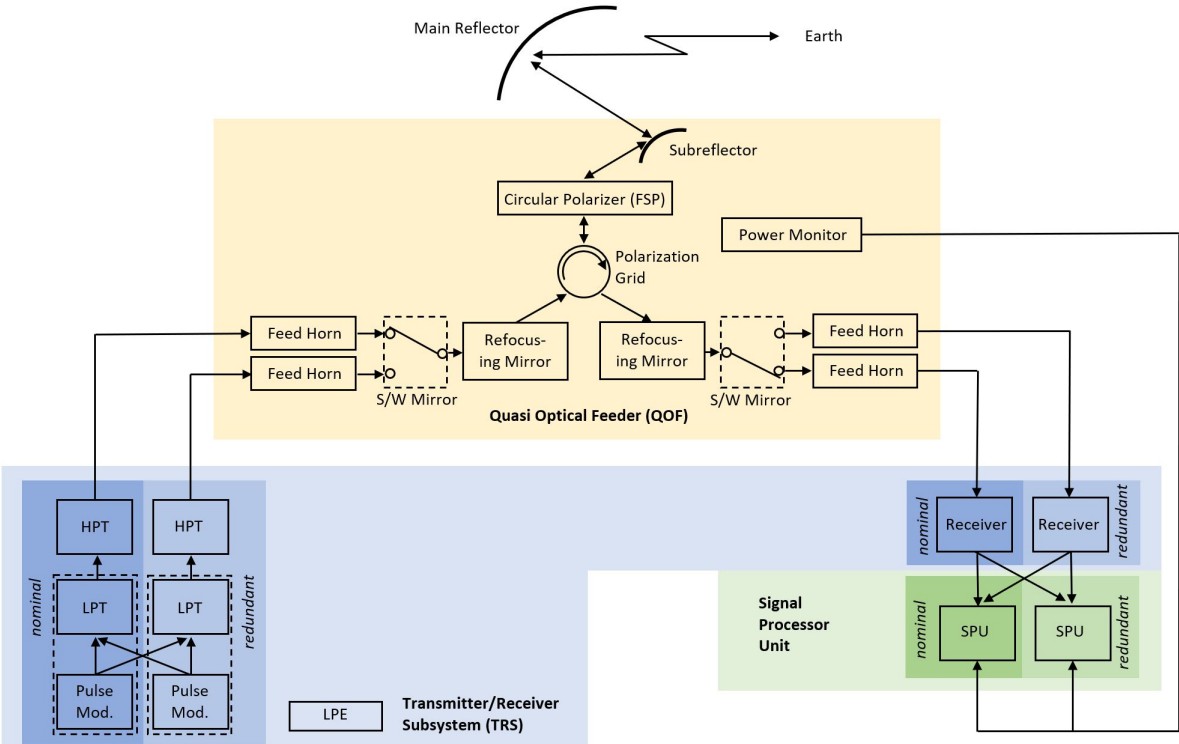

**Figure 11.** Simplified block diagram of CPR.

reducing data volume. The CPR obtains several types of calibration data to ensure and maintain performance. Specifically, the CPR is designed to acquire transmit power by using the QOF's power monitor capability. Receiver performance (e.g., NF) is determined with the noise source's signal in the RCV. During internal calibration mode, linearity and bias of the logarithmic amplifier in the SPU is acquired utilizing internal IF signal source, a step attenuator, and the data acquired by a terminated logarithmic amplifier.

The overall performance of the CPR will be evaluated using an ARC (Active Radar Calibrator) on the ground for which the CPR is configured to external calibration mode. The ARC is used to confirm the CPR transmitter performance (ARC in receiver mode) and receiver performance (ARC in transmitter mode) as well as overall performance (ARC in transponder mode with returning signal delay to avoid the contamination with ground echo). In addition, radar signal acquired by the ARC during the ARC fly-over will provide the antenna pattern and information on pulse length.

For the sea surface calibration (sea surface calibration mode), the EarthCARE satellite will perform a roll manoeuvre at regular intervals (e.g. once a month). Sigma 0 for various incident angles are useful to evaluate the CPR performance, since the normalized radar cross section of the sea surface echo (namely, sigma 0) has clear incident angle dependency and its shape is dependent on sea surface wind. In addition, other natural targets will be considered for the CPR Doppler performance evaluation.



In CPR level 1 processing, various CPR calibration (explained above) are conducted to retrieve calibrated radar reflectivity and Doppler velocity, which are used for higher level product processing. In order to derive radar reflectivity, received echo power is calibrated with calibration load in the CPR receiver. Then, the calibrated received echo power is transformed to the radar reflectivity with use of noise level estimation and CPR transmit power estimation. For the Doppler velocity product, pulse-pair covariance is calibrated with transmit RF phase change between inter-pulses (almost negligible) and the satellite velocity contamination in the CPR beam direction. Then, Doppler velocity is derived from calibrated pulse-pair covariance. CPR level 1 processing also includes surface echo bin detection, estimation of normalized radar cross section of the surface, and calculation of Doppler velocity of the surface. Those products are useful to validate geometric accuracy, radiometric accuracy, and Doppler product accuracy.

The CPR Level 1b product contains the radar reflectivity and vertical Doppler velocity (Eisinger et al., 2022).

## 5.3 The Multi-Spectral Imager - MSI

The Multi-Spectral Imager (MSI) will provide scene context information left and right of the satellite nadir track where the active instruments, ATLID and CPR, are measuring vertical cloud/aerosol/precipitation profiles. MSI observations are used to derive "MSI-only" cloud and aerosol data products across the swath (Docter et al., 2023; Hünerbein et al., 2022a, b; Nakajima et al., 2019). The MSI nadir pixel is used in the retrieval of synergistic products with ATLID (Haarig et al., 2022) or ATLID and CPR (Mason et al., 2022a; Okamoto et al., 2022). Across-track data of selected MSI channels are furthermore used for the creation of 3D scenes (Qu et al., 2022a), which are needed for 1D and 3D radiative transfer calculations to calculate radiative properties (Cole et al., 2022; Oikawa et al., 2018; Okata et al., 2017; Yamauchi et al., 2022) and perform closure assessment (Barker et al., 2022). The MSI is also used to improve the unfiltering of BBR measured radiances (Velázquez-Blázquez et al., 2022a).

The MSI is a nadir-pointing push-broom imager with a swath with of 150 km. The swath is shifted sideways and extends 115 km to the left and 35 km to the right side of nadir, in order to minimize the number of pixels affected by sun glint. (The satellite is flying southwards on the day side of the orbit and crosses the equator in the early afternoon.) The pixel sampling size is 500 m. The instrument has two cameras, a VNS (Visible, Near-Infrared (NIR), Short-wave Infrared (SWIR)) camera with four "solar channels" and a TIR (Thermal Infrared) camera with three "thermal infrared channels".

Fig. 12 shows an illustration of the MSI optical bench. The two apertures of the VNS camera and the TIR aperature are pointing towards the Earth. The TIR cold space view is used for calibration of the TIR and the sun view for the VNS calibration. The Front End Electronics (FEE) are shared between the two cameras. (They are not visible in the illustration as they are located behind the TIR camera. Also, the Instrument Control Unit (ICU) is not shown as it is enclosed within the satellite.) The optical layout of the VNS camera is shown in Fig. 13 and the TIR in Fig. 14. Table 4 describes the MSI channel definition and the radiometric performance.

The instrument has a power consumption of 60 W and a total mass of 50 kg, which is 40 kg for the optical bench module and 10 kg for the ICU. The maximum data rate is 611 kbps. SSTL (UK) is the instrument prime and manufacturer of the TIR




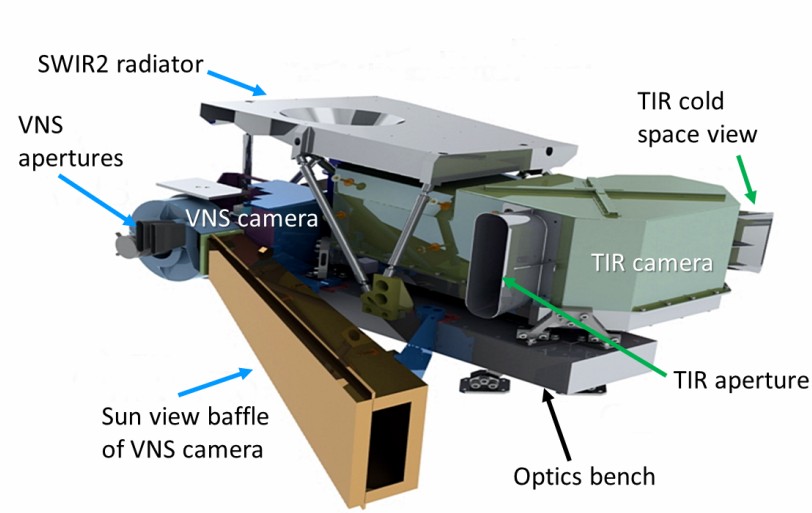

**Figure 12.** MSI optical bench. VNS stands for Visible-NIR-SWIR camera, TIR stands for Thermal-Infrared camera. (Courtesy of SSTL, UK.)

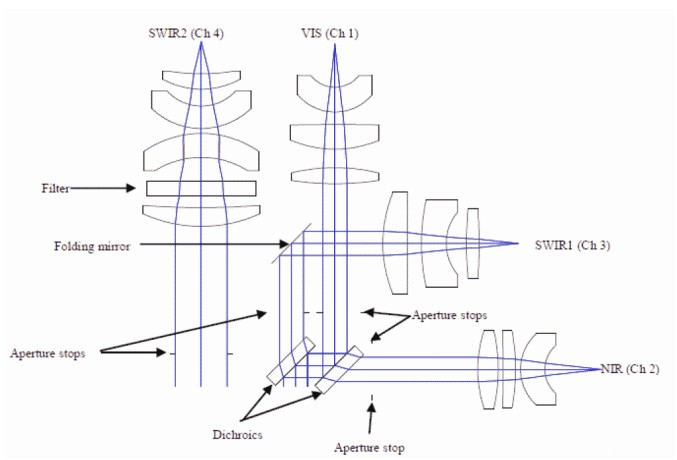

**Figure 13.** MSI VNS optical layout. (Courtesy of SSTL, UK.)

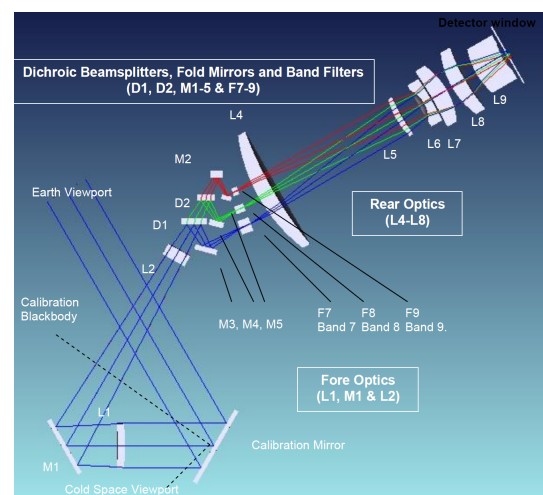

**Figure 14.** MSI TIR optical layout. (Courtesy of SSTL, UK.)

camera and FEE. The VNS camera is built by TNO (Netherlands). The Instrument Control Unit (ICU) is made by TAS (UK), the VNS detectors by XenICs (Belgium), the TIR micro-bolometer array by ULIS (France) and the blackbody by ABSL (UK).

    **The VNS camera** images light from the scene onto four different focal planes (one per channel). Each is equipped with a linear photodiode array (silicon detectors) with 512 pixels of 25 $\mu$m. The target scene illuminates 360 pixels. Bands VIS, NIR and SWIR1 share a common 4.7 mm aperture and are separated by a system of dichroics and mirrors. The SWIR2 band has a





| Channel | Centre Wavelength | Channel width | Def. Low Reference Signal* | Absolute Radiometric Accuracy | Performance SNR/NEΔT | Def. High Reference Signal* | Absolute Radiometric Accuracy | Performance SNR/NEΔT |
|---|---|---|---|---|---|---|---|---|
| Visible | 0.67 $\mu m$ | 0.02 $\mu m$ | 30 | 2.20% | 203 | 444.6 | 2.20% | 2560 |
| NIR | 0.865 $\mu m$ | 0.02 $\mu m$ | 17 | 2.20% | 137 | 282.7 | 2.20% | 1620 |
| SWIR1 | 1.65 $\mu m$ | 0.05 $\mu m$ | 1.5 | 2.20% | 26 | 67.9 | 2.20% | 1082 |
| SWIR2 | 2.21 $\mu m$ | 0.02 $\mu m$ | 0.5 | 2.20% | 138 | 24.6 | 2.20% | 5606 |
| TIR 1 | 8.8 $\mu m$ | 0.9 $\mu m$ | 220 | 0.6 K | 0.45 K | 293 | 0.38 K | 0.13 K |
| TIR 2 | 10.8 $\mu m$ | 0.9 $\mu m$ | 220 | 0.4 K | 0.25 K | 293 | 0.36 K | 0.10 K |
| TIR 3 | 12.0 $\mu m$ | 0.9 $\mu m$ | 220 | 0.54 K | 0.35 K | 293 | 0.49 K | 0.16 K |

**Table 4.** MSI performance predicted at end of life. *Units of Reference Signals: $Wm^{-2}sr^{-1}\mu m^{-1}$ (Vis-SWIR2 channels), K (for TIR channels).

larger, separate 10.4 mm aperture. Its corresponding focal plane is stabilised at 235 K via a passive cooling system. The field of view is 11.5° at f/4.58 for the VIS/NIR/SWIR1 channels and at f/2.09 for SWIR2. The optical components include glasses such as NLaSF31, ZnS and Ge in the SWIR channels. For the calibration, the camera view can be switched to a solar viewing port, where a pair of Spectralon diffusers are illuminated by the sun. For the protection of the camera and for the acquistion of the dark signal the aperture of the instrument can be closed.

A standard InGaAs detector is used for the SWIR1 channel. The SWIR2 channel uses an extended wavelength InGaAs detector, which offers better performance at high temperature operations than MCT (Mercury Cadmium Telluride). The SWIR2 detector is cooled to 235 K via a dedicated radiator. Each diode array includes two readout circuits, which allows the use of odd/even deduncancy. The use of the same read-out electronics for each detector array allows for a single design serving all four channelss, which simplifies the FEE. Multiple detector exposures are accumulated within a single ground line for improved

SNR. The pixels at the edges of each array, which are not illuminated, are used for monitoring detector thermal leakage current.

The VNS calibration unit consists of a carousel with two openings, one nadir and one sun-pointing. Inside the unit, a rotating disc can be positioned so that the nadir view is open for Earth observations (nominal operation), while the sun acquisition port is closed. Dark signal acquisition will be performed when both nadir and sun ports are closed. For solar calibration, the nadir is closed and the sun light illuminates a pair of Quasi Volume Diffuser (QVD), one for the VIS-NIR-SWIR1 aperture and

one for the SWIR2 aperture. Two pairs of QVDs are installed, which use the same optical chain to the detectors as is used in nominal operation, such that any change in the optics chain transmission can be detected and corrected in the ground processor. The first QVD pair is used for a daily radiometric calibration of VNS channels against the known scene provided by the sun. The second QVD pair is used monthly in order to monitor aging of the QVD pair that is used daily, which can be caused by contamination and solar UV exposure due to their more frequent use. The solar calibration is performed over the polar region,





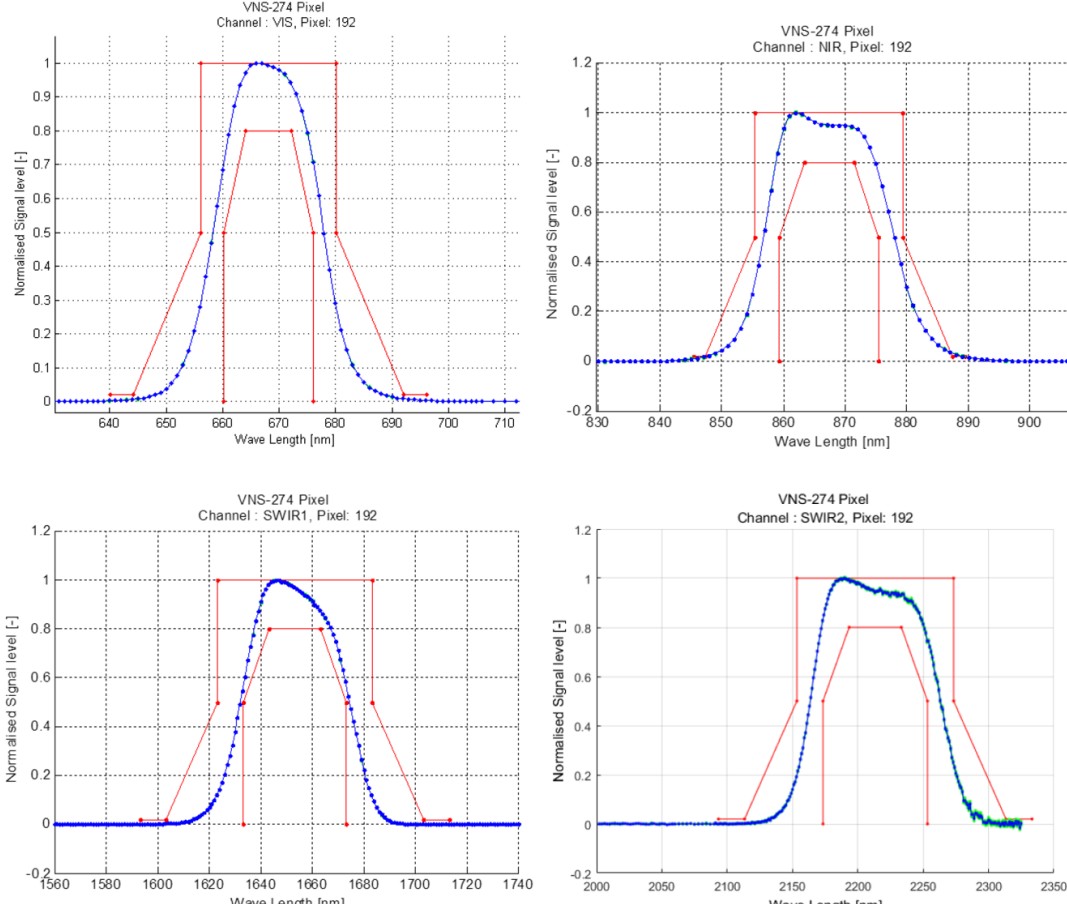

**Figure 15.** MSI VNS filter functions for the nadir pointing pixels. The red lines indicate the filter function requirements. (Courtesy of SSTL, UK.)

while the satellite is flying over the shadowed Earth with the Sun still within the line of sight. The solar aperture is baffled in order to avoid light scattered by the satellite or the solar panels.

The VNS calibration is based on

$$\Omega_g(t) = \frac{S_E(t) \cdot S_D(t')}{S_D(t) \cdot S_M(t')} \cdot \Omega_M(t_0) \tag{1}$$

where $\Omega_g(t)$ = BSDF (bidirectional scattering distribution function) of the Earth scene at time $t$, $S_E(t)$ = Earth signal at time $t$, $S_D(t)$ = signal from short-term calibration diffuser at time $t$, $S_D(t')$ = signal from short-term calibration diffuser at time $t'$, $S_M(t')$ = signal from long-term diffuser at time $t'$, $\Omega_M(t_0)$ = BSDF of long-term diffuser characterised on-ground.

The measured VNS channel filter functions are shown in Fig. 15. The VNS channel central wavelengths have a small dependency on the viewing angle, caused by bandpass filters on curved optical surfaces (lenses), resulting in an angular swath dependency of the central wavelength as shown in Fig. 17. We refer to this effect as the Spectral Misalignment Effect (SMILE).



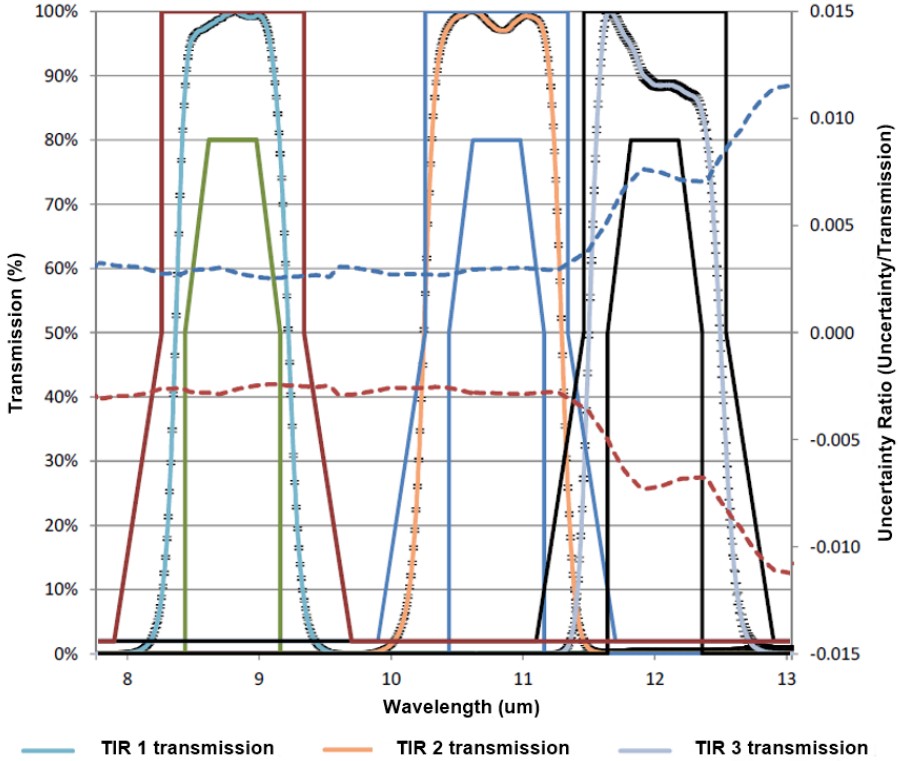

**Figure 16.** MSI TIR filter functions. The filter function requirements are indicated by the solid lines (TIR 1: red and green, TIR 2: blue, TIR 3: black). The dashed lines indicate the $3\sigma$ uncertainties on the transmission values (red: low reference signal, blue: high reference signal, according to Table 4). (Courtesy of SSTL, UK.)

(Although this is not identical to some previous use of the term "smile" in the literature, e.g., as discussed by Fisher et al. (1998).) This effect, as well as slight dependence of the point spread function on the viewing angle is considered in the MSI Level 2 retrievals.

    **The TIR camera** uses a two part imaging system, consisting of a telescope and a relay, to focus the three channels onto a single area array detector. A rotating mirror (calibration mirror) allows the line of sight to be redirected for calibration purpose onto an internal, warm blackbody and to deep space. This calibration is done once per orbit.

    The calibration mirror reflects the incoming signal (from Earth or blackbody or cold space) onto a lens (L1 in Fig. 14) that forms the aperture. The elements L1, M1, L2 create an image of the scene at long focal length. The beam then enters the rectangular Filter and Dichroic Optical Components (FDOC) where it is split by two dichroic beam-splitters (D1, D2) into three beams, which are reflected onto a common image plane and pass through their respective filters that define the response for each of the channels. An intermediate focus is formed within the FDOC, so that the filters define a field stop. The system is close to tele-centric in the region of the dichroics and filters. The filters are tilted up to $10°$ in order to control stray reflections, such that they all systematically reflect radiation from an internal reference inside the inner shroud, thereby providing control



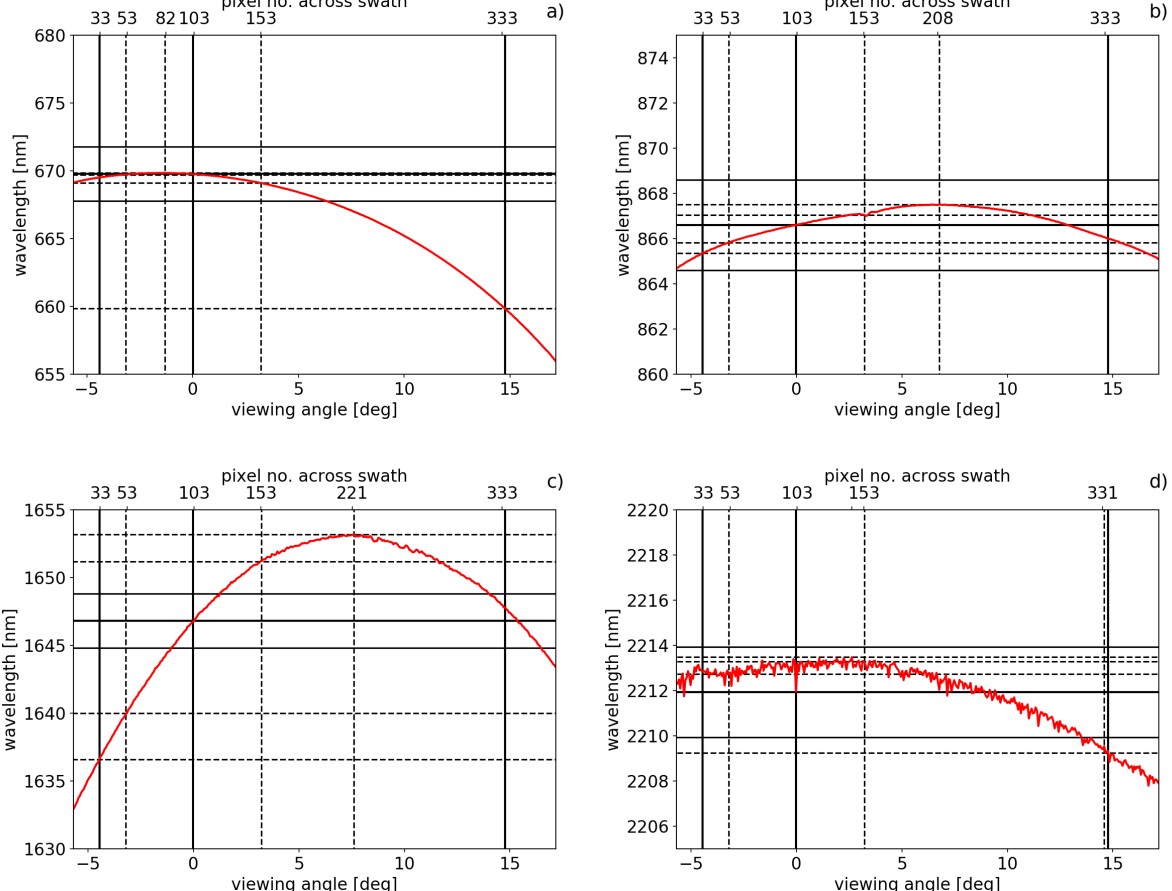

**Figure 17.** "SMILE" effect: MSI VNS central wavelength in dependence on viewing angle (bottom x-axis) and pixel number across track (top x-axis) for a) VIS, b) NIR, c) SWIR1 and d) SWIR2 channel. Solid vertical black lines indicate nadir view, -70 pixel (approx. -35 km) and +230 pixel (approx. 115 km) around nadir. Dashed vertical black lines indicate nadir view ±50 pixel (approx. ±25 km). Dashed horizontal black lines indicate corresponding central wavelength and solid horizontal grey lines indicate ±2 nm around nadir central wavelength. (Courtesy of A. Hünerbein, TROPOS, Germany.)

of the out-of-band radiation directed by the filters to the detectors. The detectors form an image of both the filters as well as the mounting frame, which acts as an internal reference body for the temperature of the rear optics structure, so that it can be
corrected for by subtraction from the scene signal.

     The beams are re-imaged onto the microbolometer array with substantial de-magnification, with the final image at approximately f/1 with a ±11.3° field of view. The rear optics is contained in a thermally isolated enclosure, controlled in temperature to better than 100 mK. The optics include Zinc Selenide, Germanium and Gallium Arsenide elements. Athermalisation is achieved by structural means; an Invar metering rod holds the detector in a position fixed with respect to the optics lens bar-
rel over a 50° temperature range. The optic surface shapes include a number of aspherical components in order to meet the



imaging requirements. The optical system operates near the diffraction limit. It uses Time Delay Integration (TDI) to improve the signal to noise ratio: signals from a predetermined number of rows ($N$) are co-added from $N$ successive frames, which effectively multiplies the total ground signal from each ground pixel by the TDI factor, $N$. MSI TIR uses a TDI factor of 19, applied on-board, which permits an improvement in SNR by a factor of almost 4.5.

The detector is a ULIS microbolometer area array detector of $385 \times 288$ pixels at 35 $\mu$m pitch. The device is mounted in a flat package with an integral thermoelectric cooler (TEC) for thermal control.

Reference area subtraction: Non-scene pixels, are also imaged onto defined areas of the detector and are called reference blocks which view the internal reference body (whose temperature follows that of the rear optics structure). Signal from these areas is used in the ground processor in a column-correction step to compensate for noise associated with the different read-out

circuits in the detector and the correction of small thermal drifts over time.

The TIR is calibrated according to

$$R_E = R_B \cdot \frac{S_E - S_C}{S_B - S_C} \tag{2}$$

where $R_E$ = radiance from Earth scene, $R_B$ = radiance from internal calibration black-body, $S_E$ = signal recorded from Earth scene, $S_C$ = signal recorded from cold space observation, $S_B$ = signal recorded from internal black-body observation.

The TIR measured filter functions are shown in Fig. 16.

The MSI Level 1b product contains the calibrated radiances for the solar channels (VNS channels) and the brightness temperatures for the thermal channels (TIR channels). The MSI Level 1c product contains the calibrated radiances and brightness temperatures remapped on a reference channel. The Level 1 products are described by Eisinger et al. (2022).

## 5.4   The Broad Band Radiometer - BBR

The core objective of the EarthCARE mission is to study how aerosols, clouds and precipitation impact radiation. While ATLID, CPR and MSI are used to derive profiles and even 3D aerosol-cloud-precipitation scenes, the Broad-Band Radiometer (BBR) measures the co-located reflected solar and emitted thermal radiation. From the unfiltered BBR radiance measurements (Velázquez-Blázquez et al., 2022a), broad-band fluxes at the top of the atmosphere are derived (Velázquez-Blázquez et al., 2022b). A key activity making use of these radiances and fluxes is the assessment of radiative closure, as described by Barker

et al. (2022), where radiative transfer calculation (1D for every scene plus 3D Monte-Carlo calculations for as many scenes as computationally affordable) is used on the aerosol-cloud-precipitation scenes retrieved from ATLID, CPR and MSI, to calculate heating rates and top-of-atmosphere fluxes and/or radiances, which are then compared to those measured by the BBR. The guiding performance requirement for the BBR is that top-at-atmosphere solar and thermal flux can be derived with an accuracy of 10 Wm$^{-2}$ for a scene size of 10 km $\times$ 10 km. Note, the performance requirements of the BBR are defined for a

ground pixel size of 10 km $\times$ 10 km, however, the technical implementation of the instrument allows for a variable integrated pixel size.

For a completely homogeneous scene, the conversion of nadir radiance to flux is straightforward, but in the general case of inhomogeneous scenes, the scene should ideally be observed simultaneously from all directions. As this is not possible with



a single satellite instrument, the BBR uses instead three fixed fields-of-views, one forward-pointing along the ground track,
one nadir and one backward-pointing, so that each scene is observed from three directions. Velázquez-Blázquez et al. (2022b)
describes how Angular Distribution Models are used to estimate the flux from these observations.

BBR has three fixed telescopes: pointed at nadir, in forward direction at 50° relative to nadir and in backward direction at
50° relative to nadir (leading to a zenith angle at the respective fore and aft pixel of 54-55°, see Fig. 2). Each view is mapped
onto a microbolometer linear array detector. For each of the three telescopes, the instrument measures the total spectrally
integrated radiation from 0.25 $\mu$m to >50 $\mu$m, alternating with the observation of the short-wave only part (0.25 $\mu$m to 4 $\mu$m),
by means of periodically switching a short-wave filter into the field of view. The individual pixel size results from the detector
array resolution (across-track direction) and the total-wave and short-wave sampling speed; the instantaneous field-of-view is
nominally 0.64 km × 0.64 km at nadir and 1.876 km (along-track) × 1.069 km (across track) for the two slanted views. In
the data processing on ground, these pixels are integrated to (nominally) 10 km × 10 km pixels, as indicated in Fig. 22. The
long-wave (LW) component is derived by subtracting the short-wave (SW) from the total-wave (TW) measurement.

The instrument requirements have all been defined with a nominal pixel size of 10 km × 10 km. However, the instrument's
concept of high-resolution sampling of total wave and short wave allows for flexibility in the integrated scene size. It is useful
for some applications to integrate scenes of approximately the same area of 100 km$^2$ – in order to maintain the signal-to-noise
performance – but in a narrower strip around the nadir with accordingly longer along-track dimension, as it is done by Qu et al.
(2022a), who is constructing 5 km × 21 km scenes for radiative transfer applications. (Note, 21 km scene length is used for
this application, rather than 20 km, because of the 7-km along-track periods of the joint retrieval grid, defined in the Level 1d
data product X-JSG (Eisinger et al., 2022).)

The BBR radiance product BM-RAD therefore includes the nominal pixel size of 10 km × 10 km as well as high resolution
data that the user can use to integrate to other desired pixel sizes.

It is important to advise users interested in the BBR radiances not to use the (not unfiltered) BBR Level 1 products (B-NOM,
B-SNG), because in these products the instrument effects have not yet been fully corrected for. Instead, the use of the BM-RAD
product (Velázquez-Blázquez et al., 2022a), which is unfiltered, is advised. The so-called unfiltering (removal of the instrument
effects) is part of the BM-RAD (Level 2) processing, where MSI scene data is (optionally) used for improved results.

**BBR instrument:** The TW channel is defined at 0.25 to >50 $\mu$m and the SW channel at <0.25 to 4.0 $\mu$m. The measured filter
response is shown at Fig. 20. The dynamic range is 0 to 550 Wm$^{-2}$sr$^{-1}$ for TW and 0 to 450 Wm$^{-2}$sr$^{-1}$ for SW. Table 5 lists
the radiometric performance against a set of reference scenes pre-defined for maximum and minimum SW and LW signals.
The radiometric stability is 0.5 % for both channels.

The three views (fore, nadir and aft) are implemented using three identical fixed mirror, dedicated telescopes imaging onto
linear array detectors. In order to produce a uniform system point spread function (PSF), achieved by summing of pixels, and
to minimise diffraction, each focus is deliberately de-focussed to create a triangular pixel response. For the oblique (fore and
aft) views less de-focus is applied, as otherwise spectral resolution would be decreased at the increased range of the targets.
The detectors used are linear, 30 pixel, microbolometer arrays from INO (Canada) (Proulx et al., 2017), onto which a special
gold- black coating has been applied and laser etched for the 0.1 mm × 0.1 mm pixels in order to increase their sensitivity to



| Parameter | Performance at min LW reference scene | Performance at min SW reference scene | Performance at max LW reference scene | Performance at max SW reference scene |
|---|---|---|---|---|
| SW radiometric accuracy (Wm$^{-2}$sr$^{-1}$) | 2.15* / 2.23** | 0.56* / 0.59** | 0.75* / 0.78** | 3.15* / 3.28** |
| LW radiometric accuracy (Wm$^{-2}$sr$^{-1}$) | 0.54* / 0.57** | 0.37* / 0.41** | 0.63* / 0.67** | 0.55* / 0.58** |

**Table 5.** BBR radiometric error performance, for the nominal spatial resolution of 10 km × 10 km and 1 km along-track horizontal sampling distance. * indicates performance for nadir view, ** for oblique view. For radiances (Table 6) derived from reference scenes with expected highest and lowest signal levels.

| Scene | SW radiance | LW radiance |
|---|---|---|
| max SW radiance | 401.82 Wm$^{-2}$sr$^{-1}$ | 35.14 Wm$^{-2}$sr$^{-1}$ |
| min LW radiance | 272.35 Wm$^{-2}$sr$^{-1}$ | 30.70 Wm$^{-2}$sr$^{-1}$ |
| min SW radiance | 23.01 Wm$^{-2}$sr$^{-1}$ | 85.40 Wm$^{-2}$sr$^{-1}$ |
| max LW radiance | 63.71 Wm$^{-2}$sr$^{-1}$ | 131.43 Wm$^{-2}$sr$^{-1}$ |

**Table 6.** Reference scenes radiance levels, computed using the instrument response and used for determining the BBR radiometric performance reported in Table 5.

thermal infrared (TIR). BBR's radiator is located approximately mid of the Optics Unit. The telescopes are mounted onto a
baseplate that is parallel to that radiator.

The linear detector arrays correspond to an area on ground of approximately 0.648 km (along-track) and ±10.2 km across track, for the nadir view, and approximately 1.88 km by ±16 km for the slanted views. The ground sampling distance between one TW to the next TW sample (or SW to SW, respectively) is 0.8 km. The reason for the large width of the slanted view ground track is, firstly, the use of identical telescopes and detector arrays for all three views and, secondly, it allows for compensation of
shifting field-of-view due to the Earth curvature without having to apply any telescope steering, by simply selecting the pixels relevant to the desired scene accordingly. Fig. 21 shows the acquisition sequence of individual SW and TW measurements. Fig. 22 indicates the resulting PSF of an integrated 10 km × 10 km pixel.

A chopper drum with four apertures – two of which house 2 mm thick, curved, quartz filters – rotates continuously around the telescopes, cycling the signal onto the detectors through SW–drum skin–TW–drum skin, etc., with a chopping scheme based
on GERB heritage (Harries et al., 2005). The drift dimension during detector integration is much smaller than the detector footprint. The nominal duration of each measurement view, 25 ms, is not long enough for the detector to reach a stable voltage and the digitized waveform of the measured view is processed to predict the asymptote of the exponential rise in the detected signal. The chopper drum rotates continuously and its speed is programmable. Around the Chopper Drum sits the Calibration Target Drum, which houses hot and cold blackbodies, a diffusor, three fold mirrors to view the diffusor from each telescope,





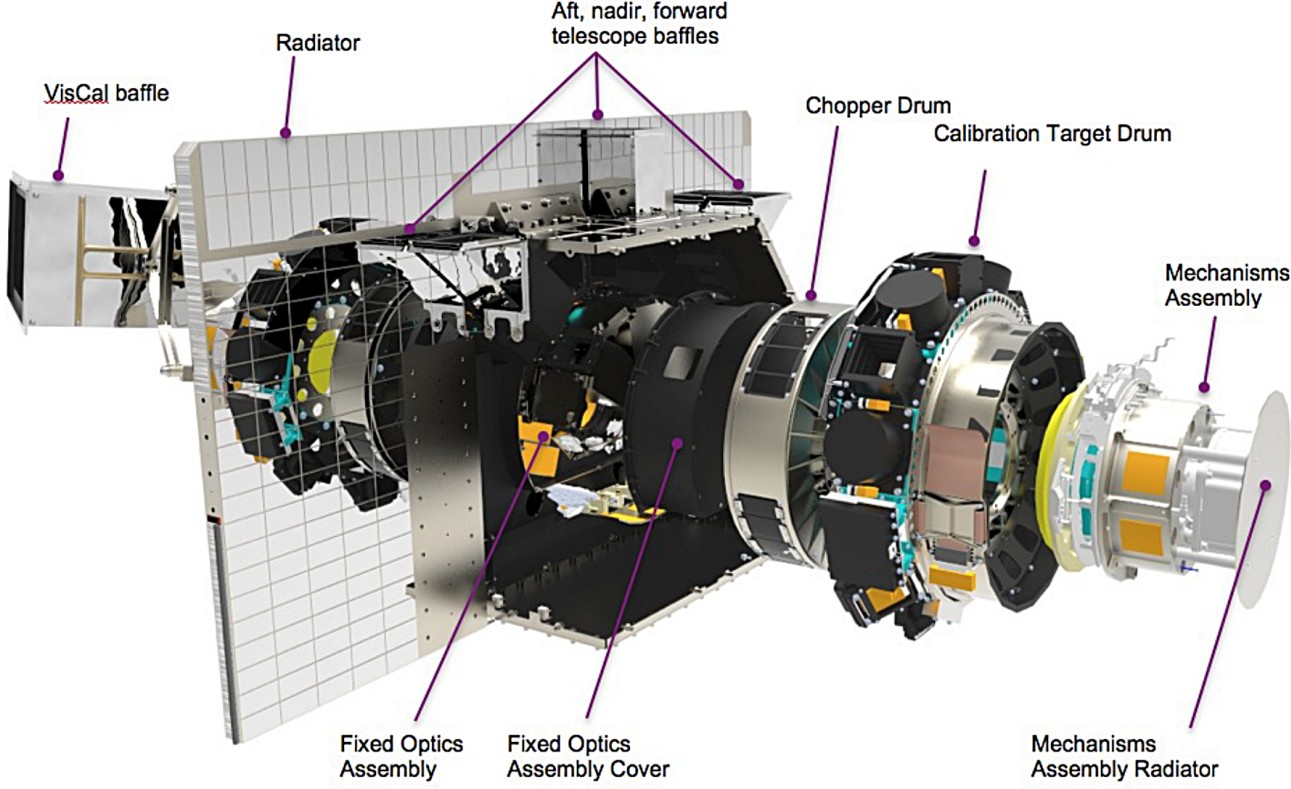

**Figure 18.** Exploded view of the BBR instrument. (Courtesy of STFC RAL Space, UK.)

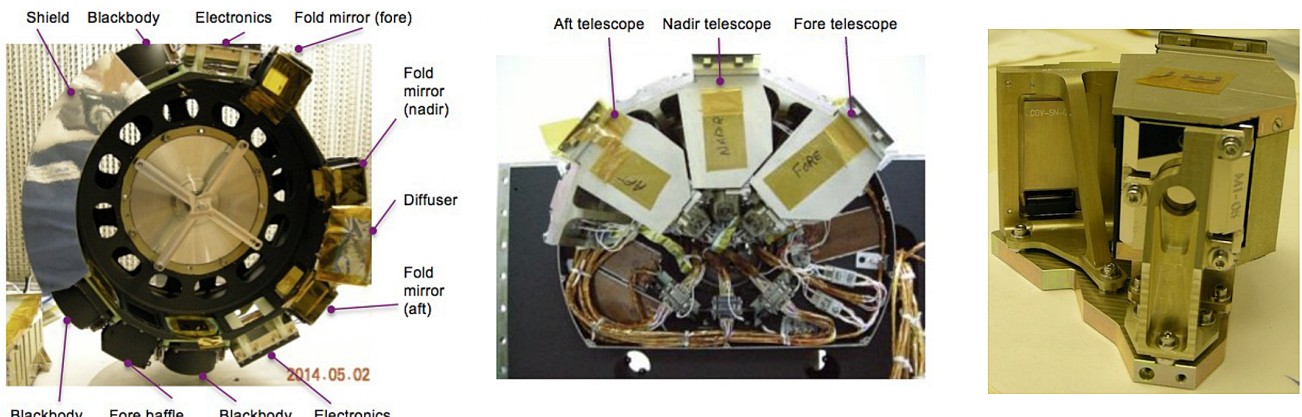

**Figure 19.** BBR Calibration Drum (left), Telescope Baseplate Assembly (centre) and photograph of one individual telescope (right). The nadir and aft baffles and one blackbody are hidden behind the shield of the Calibration Drum. (Courtesy of STFC RAL Space, UK.)



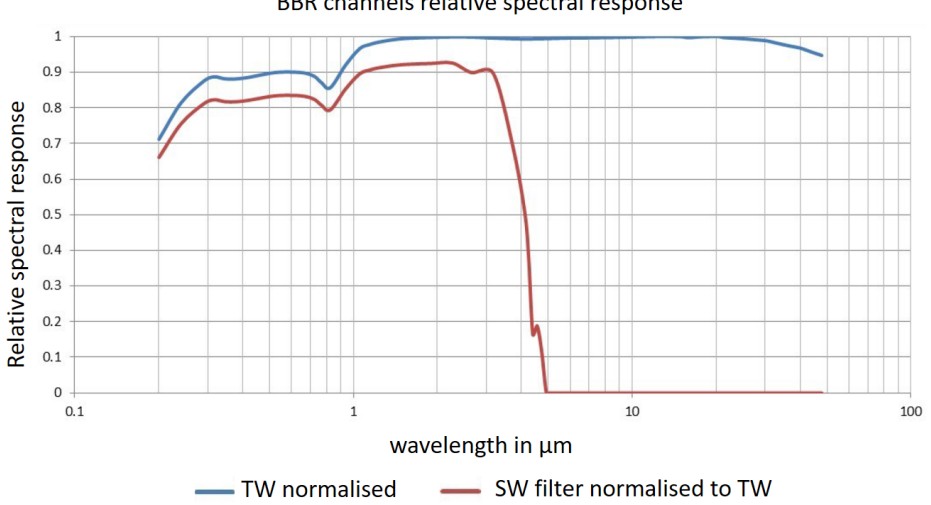

**Figure 20.** BBR TW and SW filter functions. (Courtesy of STFC RAL Space, UK.)

and the telescope baffles. The calibration target drum is rotated in steps as required. The Chopper Drum and Calibration Target Drum mechanisms are nested and located at the opposite side to the Telescope Assembly baseplate. Both mechanisms operate on dry-lubricated bearings, the chopper bearing with a self-lubricating composite PGM-HT cage and the calibration bearing with a lead bronze cage. Separate mounted boxes are located on the opposite side of the radiator and house the Signal Conditioning Electronics and the Visible Calibration (VisCal) system. The VisCal system houses a complex arrangement of

mirrors, internal baffling and an external illumination baffle, that directs solar illumination onto the Calibration Target Drum mounted diffusor.

Fig. 18 shows an exploded schematic of the instrument. Fig. 19 shows photographs of some key components of the instrument.

The BBR will perform sampling such that the views from each of the three observation angles are spatially coincident,

slightly separated in time by just over a minute between acquisitions. This requires the instrument to accommodate the residual shifts in scene position of the three fields of view that arise from Earth rotation between measurements after the implementation of yaw steering. For each telescope view a 10 km × 10 km scene Point Spread Functions (PSF) is synthesised by summation of individual pixel PSF responses along and across track. Pixels are selected to form scenes at 1 km intervals along track. To reduce the effect of overlapping radiances (from pixels outside of the nominal footprint) the PSF of edge pixels are weighted

accordingly. The match of the scenes, for the two channels and the three view directions, is controlled by the Integrated Energy (IE) requirements, Eq. 4. The IE is the ratio of the radiance measured by the instrument in a directional view $i$ and a spectral channel $j$ over a squared area of size $d$ centred around a position $(x_0, y_0)$ on Earth to the radiance measured by the instrument from the entire large and spatially uniform scene, according to



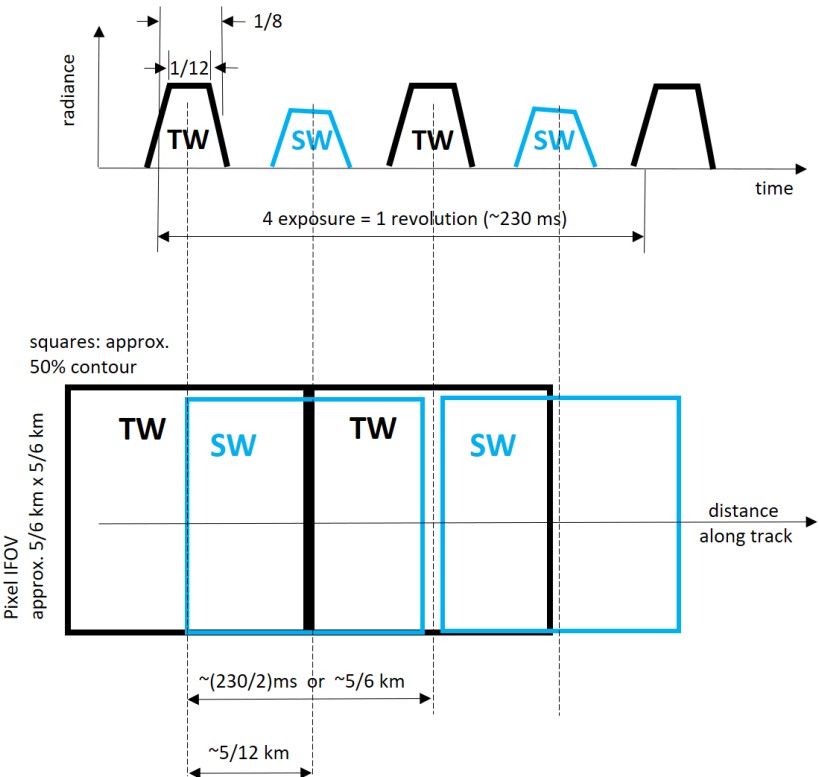

**Figure 21.** BBR: Schematic of the SW and TW acquisition sequence. This is for the nominal Chopper Drum speed. However, in order to maximise the life time of the Chopper Drum, it is considered to operate the Chopper Drum at 75% of the nominal speed, which leads to a respective stretch of the sequences. The resulting performance of in integrated 10 km × 10 km pixel will remain within required specification. (Courtesy of Thales Alenia Space, UK.)

$$IE_{ij}(d) = \frac{\int_0^\infty \iint_{y_0-\frac{d}{2}}^{y_0+\frac{d}{2}} {}_{x_0-\frac{d}{2}}^{x_0+\frac{d}{2}} PSF_{ij}(x,y,\lambda) \cdot L_i(\lambda) \cdot R_{ij}(\lambda) dx dy d\lambda}{\int_0^\infty \iint_{-\infty}^{\infty} PSF_{ij}(x,y,\lambda) \cdot L_i(\lambda) \cdot R_{ij}(\lambda) dx dy d\lambda}, \tag{3}$$

where $i$ stands for the instrument along-track directional view (nadir, forward, backward); $j$ stands for the spectral channel (SW, TW), $R_{ij}(\lambda)$ is the instrument spectral response function in directional view $i$ and spectral channel $j$, $PSF_{ij}$ is the instrument point spread function in directional view $i$ and spectral channel $j$. The IE of a ground scene in the SW and TW, respectively, over a square target of the size $\Delta X$, centred around the system PSF barycentre of the nadir view, considering the total radiance of the reference scene, is required to be identical within 5% (goal 2%) for the three directional views, according

to the Integrated Energy requirement of the nominal 10 km × 10 km pixel:



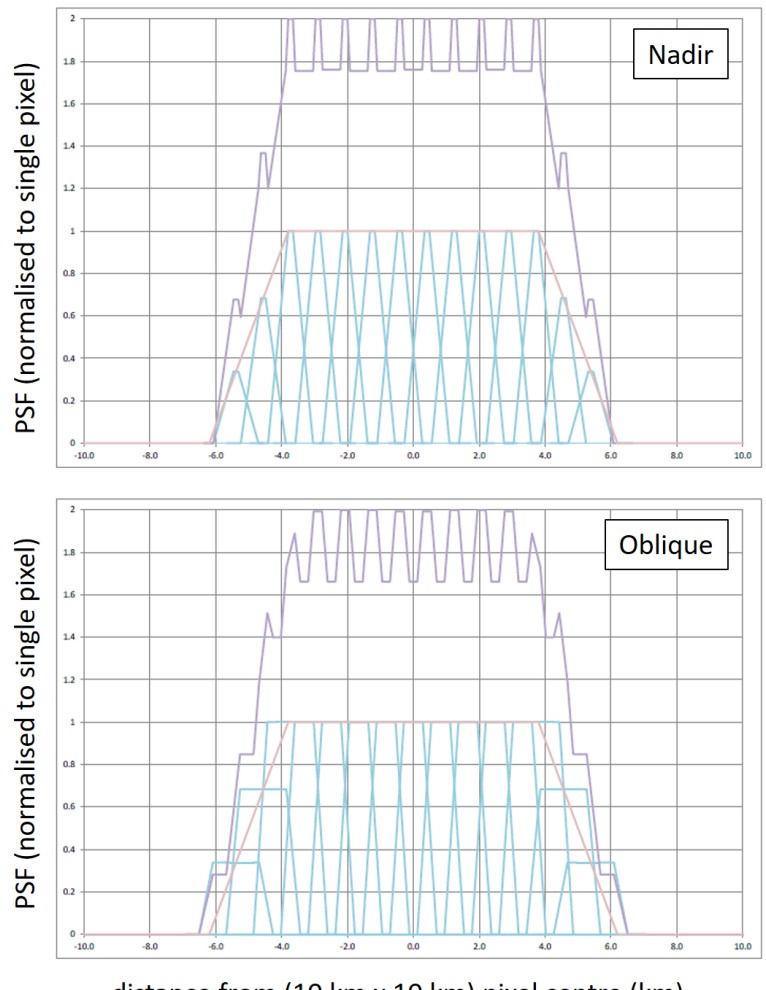

distance from (10 km x 10 km) pixel centre (km)

**Figure 22.** BBR: integration of a nominal 10 km × 10 km ground pixel from individual SW or TW, respectively, measurements. Blue lines indicate individual single pixels acquired when the chopper drum SW or TW, respectively, window was open, weighted according to the IE requirement (Eq. 4). The violet line shows the integrated 10 km × 10 km pixel PSF. This is for the nominal Chopper Speed (see comment in Fig. 21). (Courtesy of Thales Alenia Space, UK.)

$$\frac{|IE_{nadir,j} - IE_{k,j}|}{IE_{nadir,j}} \leq 0.05 \text{ (goal 0.02)}, \tag{4}$$

where $k$ stands for the forward and the backward view.

**BBR calibration** relies on the use of blackbodies. LW calibration: Gain and offsets in the measured voltage signals are removed using hot and cold blackbodies that are autonomously rotated into the field of view every 88 s. This calibration





| Parameter | Orbit Value (Mean Kepler) |
|---|---|
| Semi-major axis | a = 6771.28 km |
| Eccentricity | e = 0.001283 |
| Inclination (sun-synchronous) | i = 97.050° |
| Argument of perigee | $\omega = 90°$ |
| Mean Local Solar Time, Descending Node | MLST = 14:00 |
| Repeat cycle / cycle length | 25 days, 389 orbits |
| Orbital duration | 5552.7 s |
| Mean Spherical Altitude | 393.14 km |
| Minimum Geodetic Altitude | 398.4 km |
| Maximum Geodetic Altitude | 426.0 km |
| Average Geodetic Altitude | 408.3 km |

**Table 7.** EarthCARE Orbit Parameters.

frequency is decided by the temperature stability of the Chopper Drum skin and can be modified, if necessary, on orbit. The four instrument blackbodies are operated in hot redundancy and have been characterised on ground against a blackbody that is traceable to radiometric standards. Every six months on-orbit the blackbody temperatures are swapped, allowing a check on instrument linearity by recording data over a range of radiance.

SW calibration: LW gain is transferable to the SW, with knowledge of the filter spectral response that has been characterised
on ground. SW gain is monitored in-flight via measurements made with the VisCal system, to detect changes in instrument response due to, for example, aging. The wall of each telescope baffle incorporates a set of three Monitor Photo Diodes, in order to monitor in turn aging of the VisCal optical path. The solar calibration is performed over approximately 30 orbits every two months.

The PSF of the pixels in the microbolometer arrays are also characterised on-ground, in order to formulate the algorithm for
summing pixels to form the 10 km synthesised scene-PSF.

The BBR Level 1b product (Eisinger et al., 2022) contains in particular calibrated irradiances in various spatial resolutions. Instruments effects have not been removed from the Level 1b product. The so-called unfiltering process is taking place in the Level 2 processing, therefore, the science data user should use the BM-RAD Level 2 data product.

## 6  EarthCARE Orbit

The orbit of the EarthCARE satellite is sun-synchronous, with a descending node crossing time of 13:45–14:00 hours at the equator, an inclination of ≈97° and revisit time of 25 days (389 orbits). Table 7 shows the key parameters of the EarthCARE orbit.





Immediately after launch, a record of the orbits completed by the satellite is maintained through the use of an orbit number, starting with the first orbit and incrementing. Orbit numbers are used for various data headers and file naming (e.g. for science data) to provide an unambiguous reference to the orbit within which the data was collected. The start/end of a particular orbit, and therefore the increment of the orbit count, is taken to be the Ascending Node Crossing (ANX) (i.e. one complete EarthCARE orbit starts from one ANX and ends at the next ANX). Orbit number 1 for the EarthCARE mission begins at the first Ascending Node Crossing (ANX) after launcher separation.

The orbit parameters according to Table 7 are actively maintained, in particular, to compensate for atmospheric drag. On-board thrusters are used for orbit maintenance manoeuvres, so-called "$\Delta v$" manoeuvres. (This name hints to the use of the satellite on-board thrusters which will be used to achieve a difference, $\Delta$, in the satellite velocity, $v$.)

The baseline $\Delta v$ manoeuvre (so-called 0-degree in-plane manoeuvre, meaning, the spacecraft is accelerated along its flight vector in the orbital plane) is used for ground-track and attitude maintenance through semi-major axis and eccentricity control. Depending on the level of solar activity, it must be carried out (around the equator) every 10 days (high solar activity) or, in the best case of low solar activity, every 30 days. The full operation takes 2000 s outside nominal operation, while the instruments are switched down for safety, with a maximum period of 600 s operations of the thrusters. As an alternative, small $\Delta v$ manoeuvres might be used, which are of much shorter maximum duration, allowing the spacecraft to stay in nominal pointing mode and do not require the instruments to switch off. In this case, the attitude disturbance lasts only a few seconds, while the instrument science data acquisition continues. This operation lasts in total about 500 s, with a thruster burn duration of approximately 3 s, leading to a total out-of-specification pointing period of about 5 s, which is flagged in the data products accordingly. The small $\Delta v$ manoeuvres would be grouped together on consecutive orbits during approximately one week with many small $\Delta v$ manoeuvres (10 to 90 pairs per week). Both baseline and small $\Delta v$ manoeuvres will be tested during the Commissioning Phase (see Section 7) in order to identify the preferred baseline for routine orbit maintenance operations.

There are also two other manoeuvres: (1) 90-degree, out of plane manoeuvre (meaning, the spacecraft is accelerated perpendicular to its flight vector in the orbital plane) which is used to control inclination and mean local solar time. It can only be performed in ascending node (eclipse), within $\pm 1$ month of equinox. There are two strategies proposed, either a single, large manoeuvre at mission midpoint or smaller manoeuvres at every equinox, which uses more fuel, but slightly increases the accuracy to which the mean local solar time is maintained as well as reduces the size of each individual manoeuvre. (2) 180-degree out of plane retrograde manoeuvre, in order to lower the orbit. Meaning, the satellite velocity is reduced, while keeping the flight vector in the orbital plane. As EarthCARE has no forward-pointing thrusters, the satellite will be rotated by $180°$, so that it is flying backwards for the duration of the manoeuvre. The only foreseen possible use of this manoeuvre is for a collision avoidance emergency.

The 0-degree in-plane $\Delta v$ manoeuvres will maintain the satellite orbit so that its ground track will stay within the so-called dead-band, which is the required accuracy of the predictable on-ground orbit track and has been set for EarthCARE to $\pm 25$ km with respect to the nominal orbit nadir track. This means, that a predicted overpass over a reference ground location has an uncertainty of $\pm 25$ km.





The satellite altitude will be maintained within $\pm 2$ km (worst case altitude variation is $\pm 2.5$ km). The exact orbit parameters, however, are always available in the Level 1 and Level 2 data products.

The ESA Fligh Operations Segment (see Section 8.1) generates daily a new predicted orbit file that covers the next seven
days, listing orbit state vectors. FOS also produces daily a reconstituted orbit file that contains the last 24 hours of actual orbit state vectors. Operators of ground-based observatories or research aircraft can use these data for their planning of measurement activities coinciding with an EarthCARE overflight, for example, for the purpose of collecting ground-based data complementary to the EarthCARE satellite observations or for EarthCARE product validation.

## 7 Mission Phases

The mission will be broken down in to the following operational phases:

1. Launch and Early Orbit Phase (LEOP), including launch, S-band communication acquisition, orbit acquisition and deployment of solar panel and CPR antenna.

2. Commissioning Phase (also referred to as Phase E1), including platform functional check (satellite basic functions and health verification), ground segment acquisition (including X-band communication), instruments switch-on and
functional/health verification, in-orbit verification of Level 1 and Level 2 performance, specific operations for calibration/validation campaign and Level 1 and Level 2 processors maintenance and evolution activities. The validation of the satellite and its instruments' performance will start during the Commissioning Phase. The Commissioning Phase is expected to be completed six months after launch.

3. Operational Phase (also referred to as Phase E2), including science data acquisition, data (re-/)processing and distribu-
tion, continuation of data product monitoring and validation, Level 2 processor maintenance and evolution and satellite and instrument calibration maintenance. This phase is expected to last at least two and a half years, with sufficient consumables margin for one additional year. With the start of this phase, Level 1 and Level 2 data products are expected to become publicly available.

4. End of Mission (also referred to Phase F). During this phase, the satellite will be no longer operational, however, ground
segment activities, such as reprocessing, may still continue.

## 8 Ground Segment and Data Products

### 8.1 Ground Segment Overview

The EarthCARE System consists of the EarthCARE satellite (Space Segment), the Ground Segment at ESA and JAXA, the interfaces to the EarthCARE users and ECMWF as the provider of weather forecast data (auxiliary data product X-MET).
ECMWF also receives EarthCARE data for quality monitoring and assimilation.



The EarthCARE Ground Segment is composed of

1. The ESA Flight Operations Segment (FOS), located in ESA-ESOC, Darmstadt, Germany. It is in particular responsible for control and manoeuvring of the satellite, commanding and health monitoring of the satellite and the instruments, mission planning, orbit (control, determination, prediction), on-board software management, and dissemination and archiving of House Keeping Telemetry (HKTM) data.

2. The ESA Payload Data Ground Segment (PDGS), located in ESA-ESRIN, Frascati, Italy. It is in particular responsible for the acquisition of science data, the generation of Level 0, Level 1 (except for CPR) and ESA Level 2 data products, the archiving of data products and auxiliary data, the user services, the distribution of Level 1 and Level 2 data products to users, the monitoring of the performance of the instruments, as well as the overall mission performance, the provision of diagnostic data to the algorithm developers and validation teams, the maintenance of the instruments' configuration settings, the calibration of the instruments, and the mission planning.

3. The JAXA Satellite Applications and Operations Center (SAOC), located in Tsukuba, Japan, is in particular responsible for the reception of CPR Level 0 data and related aux files received from PDGS and the production of the CPR Level 1 data product, the support to ESA for CPR operations, CPR performance monitoring, CPR calibration and instrument parameter updates as required, the reception on processing of other Level 1 (and Level 2) products from ESA, the user services, and the distribution of Level 1 and Level 2 data products to JAXA users.

The S-Band station, at Kiruna, Sweden, will downlink the Telemetry (TM) data from the satellite and uplink the Telecommand (TC) data. The two X-Band stations for transmission of the science data will be located in Kiruna, Sweden, and Inuvik, Canada.

Fig. 23 illustrates the elements and data flow of the overall EarthCARE Ground Segment structure. Auxiliary meteorological data are provided by ECMWF.

Fig. 24 shows the components of JAXA's EarthCARE mission operation system. It consists of data processing system (DPS), data transfer and management system (E-XING), consolidated data dissemination system (G-Portal) and 3rd-generation JAXA Supercomputer System (JSS3) used for product reprocessing. DPS receives CPR L0, L1 of ESA sensors, and other data necessary for data processing from ESA, and generates CPR L1b and higher-level products. E-XING transfers relevant data between DPS and facilities inside and outside of JAXA, and archives the data in the dedicated storage system. G-Portal disseminates JAXA and ESA products on the Internet, providing users with functions of catalogue search and data retrieval. The reprocessing of products after algorithm update is performed on the Linux server cluster deployed in JSS3.

**Data Latency:** The EarthCARE Ground Segment has been designed for fast data processing to prevent data accumulation and backlog. All instruments' Level 1b/c and Level 2a data products will be made available to users within 24 hours after sensing, Level 2b products will be made available to users within 48 hours after sensing, and any contingency case will be recovered within 5 days.



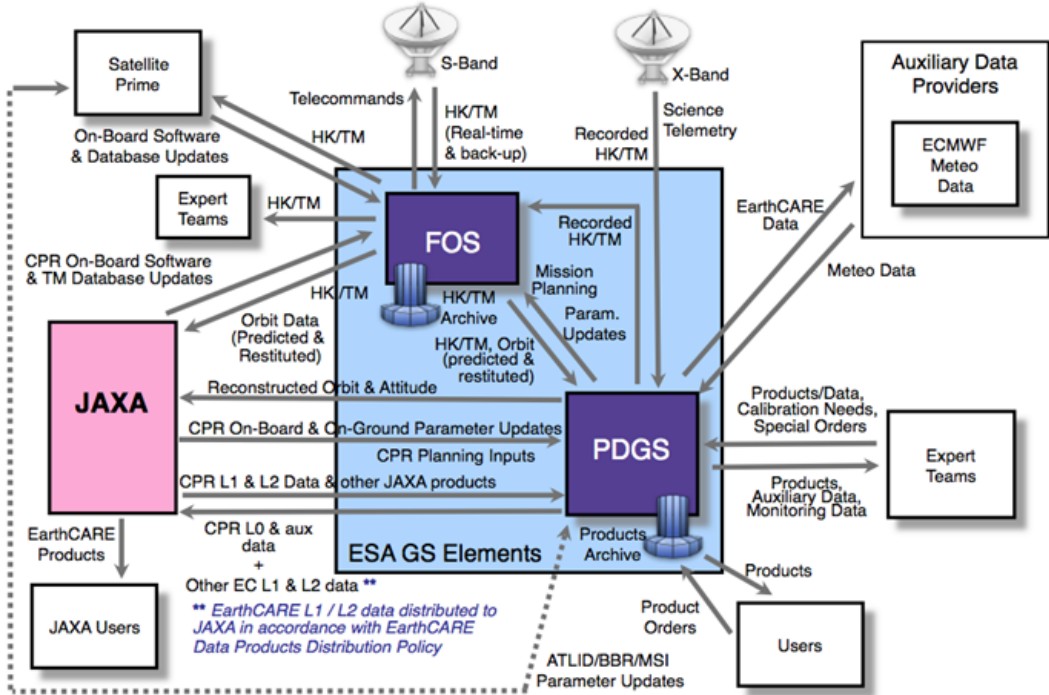

**Figure 23.** EarthCARE Ground Segment Functional Overview.

## 8.2 Data Products

All data products generated in frame of the EarthCARE mission are categorized per Product Level. The definition of each
665   Product Level has been specifically tuned for the EarthCARE mission and can be found in Table 8.

L1 products of an instrument of EarthCARE will be developed by the agency responsible for the instrument, namely L1
products of ATLID, MSI and BBR are developed by ESA and CPR L1 product is produced by JAXA. As for L2 products, both
agencies develop algorithms independently, although continuous exchange of information is being conducted between Japan
and Europe through the Joint Algorithm Development Endeavor (JADE) under the framework of the Joint Mission Advisory
670   Group (JMAG). The production model and L2 data products of ESA and JAXA are described by Eisinger et al. (2022).

Users can acquire products of both agencies from the websites of both agencies; ESA website provides JAXA products in
addition to ESA products, and JAXA G-Portal website distributes ESA and JAXA products as well. Therefore, users can access
all the products in a single place.

L0 data products consist of re-sorted science packages of the individual instruments transferred from the satellite to the
675   ground segment. They are not available to users and therefore not described. The ESA PDGS processes the L0 data products
of ATLID, MSI and BBR into the calibrated instrument Level 1b (L1b) data products. The JAXA PDGS does the same for the
CPR data products. Level 1a data products are not produced.



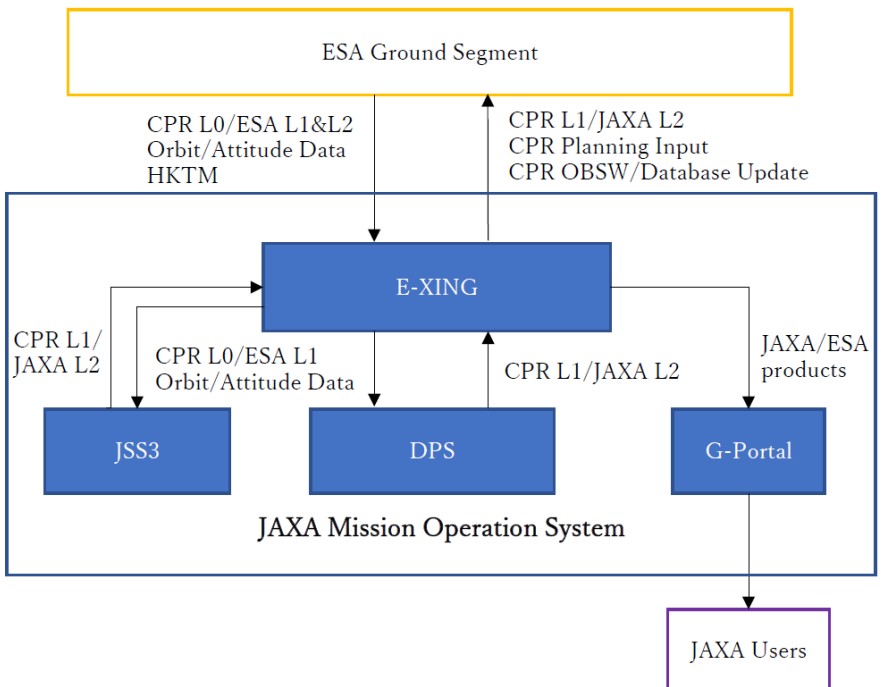

**Figure 24.** Components of JAXA mission operation system.

| Level 0 (L0) Product | Raw instrument science packets, ordered in time, with duplicates removed, annotated with quality flags and time stamps related to the data acquisition at the ground station. For expert users only, not distributed. |
|---|---|
| Level 1b (L1b) Product | Calibrated instrument data processed to physical units, with error bars, quality flags and geolocations. |
| Level 1c (L1c) Product | MSI only: L1b data re-sampled onto the grid of one selected MSI reference channel. |
| Level 1d (L1d) Product | Special/auxiliary products created to support higher-level processing of EarthCARE products. (The only L1d product is the "joint standard grid".) |
| Level 2 (L2) Product | Derived geophysical variables, either at the same resolution and location as L1b data ("native grid") or re-sampled to a common grid ("joint standard grid"), with error bars, quality flags and geolocations. |
| Level 2a (L2a) Product | (EarthCARE specific definition:) L2 product derived from one single EarthCARE instrument. |
| Level 2b (L2b) Product | (EarthCARE specific definition:) L2 product synergistically derived from two or more EarthCARE instruments. |

**Table 8.** EarthCARE Data Product Levels.





The L1b products are calibrated and geolocated instrument data in scientific units and consist of the data products C-NOM, A-NOM, M-NOM, B-NOM, B-SNG, plus additional calibration products, which are generally not of interest to scientific users and therefore not described here. M-RGR is a re-gridded MSI L1c product. (Note, users interested in BBR radiances are advised to use BM-RAD instead of B-NOM or B-SNG.) As the only L1d data product, the X-JSG data product, contains the "Joint Standard Grid" used by many of the L2a and L2b processors as a retrieval grid. This is required in particular for synergistic retrievals as every instrument has its own characteristic spatial sampling, while for the geophysical product retrieval, mapping of the various input data onto a joint grid is a prerequisite. The L1b/c/d data products are described by Eisinger et al. (2022).

The L2 data products include a comprehensive range of geophysical parameters related to aerosols, clouds, precipitation and radiation. Fig. 25 and Fig 26 provide an overview of the ESA and JAXA L2a/L2b data products containing retrieved aerosol, cloud, precipitation and radiation parameters.

A complete list of all L2a and L2b data products with references to the corresponding publications describing the products, retrieval algorithms and their validation is given by Eisinger et al. (2022).

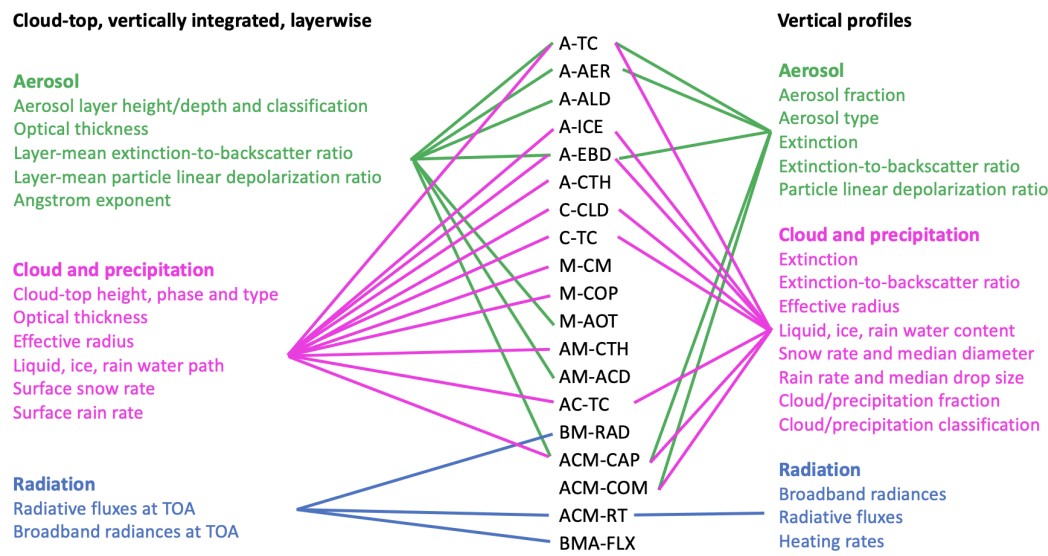

**Figure 25.** Overview of ESA L2a and L2b data products containing retrieved aerosol, cloud, precipitation and radiation parameters. The column in the middle lists the names of the respective L2 data products.

## 8.3 Level 2 Retrievals Supporting Science

For preparation and verification of the scientific processors that are producing the L2a and L2b data products, a number of supporting scientific activities have been carried out. In particular, the development of an EarthCARE end-to-end simulator and suitable test scenes was required. Furthermore, an aerosol model as a basis for aerosol typing was developed and a study was



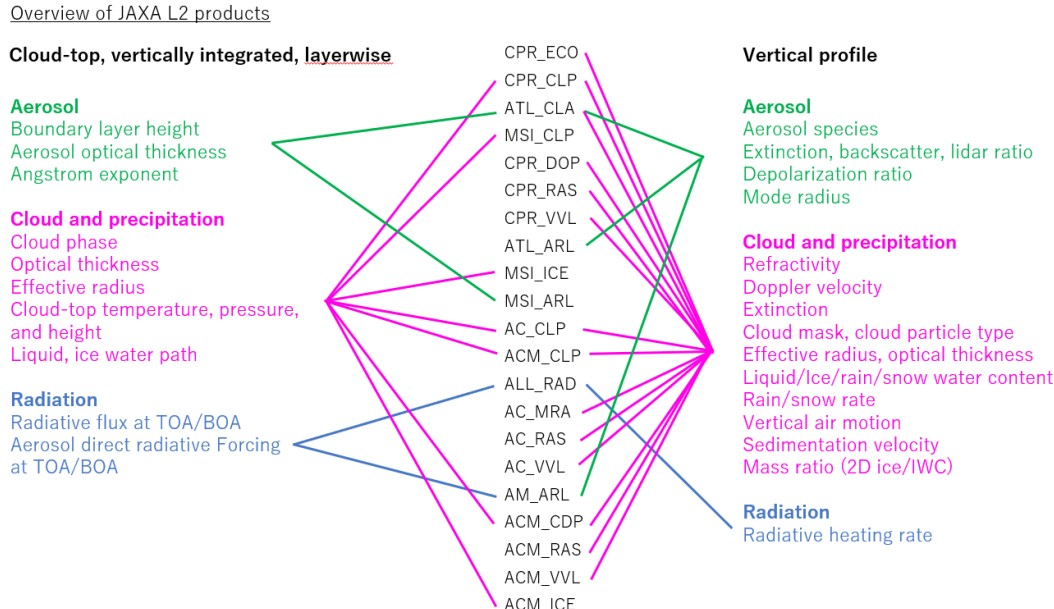

**Figure 26.** Overview of JAXA L2a and L2b data products containing retrieved aerosol, cloud, precipitation and radiation parameters. The column in the middle lists the names of the respective L2 data products.

carried out to inter-compare the microphysical assumptions used in the cloud retrievals in the different frequency/wavelength domains applicable to EarthCARE observations.

To this end, the EarthCARE end-to-end simulator (E3SIM) has been developed to simulate EarthCARE observations and provide a development and test environment for L1 and L2 processors (Eisinger et al., 2022).

For the purpose of algorithm development and testing, test scenes have been generated using the Canadian GEM model (Qu et al., 2022b). These test scenes have been processed by E3SIM to generate simulated CPR, ATLID, MSI and BBR L1b data products (Donovan et al., 2022a).

To stimulate algorithm development in Japan, JAXA has been developing Joint Simulator for Satellite Sensors (Joint-Simulator) which is a suite of software that simulates EarthCARE observations using numerical model data as input (Hashino et al., 2013, 2016). The basic structure of Joint-Simulator is inherited from Satellite Data Simulator Unit (SDSU; Masunaga et al. (2010)) and NASA Goddard SDSU (Matsui et al., 2013). The EarthCARE sensors are simulated by EarthCARE Active Sensor simulator, EASE, (Okamoto et al., 2003, 2007, 2008; Nishizawa et al., 2008) for the Doppler CPR and ATLID, RSTAR 6b Nakajima and Tanaka (1986, 1988) for MSI and MSTRN X (Sekiguchi and Nakajima, 2008) for BBR. The main purpose of Joint-Simulator is to simulate the signals from EarthCARE's four sensors within the model for validating the aerosol and cloud fields simulated by Cloud Resolving Model (CRM) and General Circulation Model (GCM) after the launch. The Joint-Simulator has also been used with a secondary purpose for algorithm evaluation before the launch. Here, the Joint-Simulator is applied to a Japanese Global Cloud Resolving Model (GCM), Nonhydrostatic ICosahedral Atmospheric Model (NICAM)



and produces EarthCARE Level 1 test products that are then used for preparing EarthCARE algorithms after the launch. For example, Hagihara et al. (2021) studied Doppler velocity simulations from the Joint-Simulator and showed the standard deviation of random error in the cirrus clouds case for the Doppler simulation was decreased to 0.5 m/s for -10 dB Ze after 10-km horizontal integration. Hagihara et al. (2022) showed another study on a global assessment of Doppler velocity errors of EarthCARE/CPR using the NICAM and the Joint-simulator. The MSI observation data is expected to be unique where its center wavelength has a dependency on its position in the across-track observation swath ("SMILE" effect, see section 5.3). Wang et al. (2022) investigated the impact of this MSI SMILE effect on the cloud properties using the simulated L1b data by the Joint-Simulator.

For aerosol retrievals, a common baseline for the development, evaluation and implementation of EarthCARE algorithms is required to ensure consistency of different aerosol products from the multi-instrument platform and to facilitate the conform specification of broad-band optical properties needed for the EarthCARE radiative closure efforts. This hybrid end-to-end aerosol classification model is described by Wandinger et al. (2022a).

A range of L2a and L2b data products is available with retrieved aerosol, cloud and precipitation quantities from one or several of the EarthCARE instruments, exploiting the fact that CPR, ATLID and MSI are operating in wavelength or frequency regions that are very different from each other.

Fourteen ESA L2a and ten L2b products are based on individually optimised retrieval algorithms for each one of the Earth-CARE instruments or combination of instruments for synergistic exploitation. Different instruments have different and specific sensitivities to the various atmospheric parameters of interest and retrieval algorithms must use specific assumptions, in particular, related to microphysics and optical properties. Mason et al. (2022b) provides an inter-comparison overview of the different cloud, aerosol and precipitation retrievals of the European/Canadian L2a and L2b data products.

| Paper content | Reference |
|---|---|
| EarthCARE end-to-end simulator E3SIM | Eisinger et al. (2022) |
| GEM test scenes for L2 development and testing | Qu et al. (2022b) |
| Generation of simulated observations from test scenes using E3SIM | Donovan et al. (2022a) |
| Aerosol classification model | Wandinger et al. (2022a) |
| Inter-comparison of retrievals of different processors | Mason et al. (2022b) |
| Joint-Simulator | Hashino et al. (2013, 2016) |
| Generation of simulated observation using Joint-Simulator | Roh et al. (2022) |

**Table 9.** Important supporting references related to L2 products and algorithms.

After the Commissioning Phase, expected to last six months after launch, there will be full and open access to the EarthCARE L1 and L2 data sets, free of charge, provided online via ESA and JAXA websites. Access to data is granted upon electronic online registration, with the user accepting the Terms and Conditions. The Terms and Conditions cover all types of use. No



specific proposal describing the use of the data needs to be submitted. During the Commissioning Phase, data access is limited

to the accepted participants in the validation program.

## 9 Conclusions

The EarthCARE mission is implemented by ESA in cooperation with JAXA to acquire key observables needed to better understand the role of clouds, aerosol and radiation on climate. The satellite platform and three instruments, the cloud-aerosol lidar instrument ATLID, the imager MSI and the radiometer BBR, are provided by ESA, the cloud Doppler radar CPR is

provided by JAXA. All instruments have been characterised and their required performances have been confirmed. The satellite will fly in a sun-synchronous polar orbit with a descending node at 14:00 hours mean local solar time at the equator. The EarthCARE Ground Segment will operationally produce a large number of data products, including comprehensive sets of geophysical data related to aerosol, cloud, precipitation, vertical motion of cloud particles, together with corresponding heating profiles, broad-band radiative fluxes and radiances. The mission will be launched in 2024.

*Author contributions.* D. Bernaerts, the ESA EarthCARE Project Manager, and T. Tomita, the JAXA EarthCARE CPR Project Manager, provided the overall guidance on the technical and programmatic context. K. Wallace contributed in particular to the MSI and BBR instrument and performance descriptions. G. Tzeremes contributed to the ATLID instrument and performance description. H. Nakatsuka and Y. Ohno contributed to the CPR instrument and performance description. P. Deghaye contributed to the description of the ESA Ground Segment. M. Taga contributed to the description of the JAXA Ground Segment. M. Eisinger ensured the consistency with (Eisinger et al., 2022). R.

Koopman, S. Rusli M. Kikuchi, and T. Tanaka contributed to the science background and data product overview. T. Wehr and T. Kubota prepared the manuscript with contributions from all co-authors.

*Competing interests.* The authors declare that they have no conflict of interest.

*Acknowledgements.* We would like to thank our industrial partners working on the satellite and payload, including the satellite prime, Airbus Defence and Space (Germany), the base platform responsible, Airbus Defence and Space (UK) and the instrument teams at NEC (Japan),

SSTL (UK), TNO (The Netherlands), Airbus Defence and Space (France), Selex Galileo (Italy), STFC RAL Space (UK), SSTL (UK), Thales Alenia Space (Switzerland), and LEONARDO (Italy). Sincere thanks also to the members of the Joint Mission Advisory Group that advises ESA and JAXA on science aspects of the mission, and the European, Canadian and Japanese members of the Level 2 development teams and ECMWF, as well as the long standing support and advice from the Principal Investigators and Project Scientists of NASA missions CloudSat and CALIPSO and the CERES instrument. We would also like to thank our colleagues and members of the ESA and JAXA teams and NICT.

A special acknowledgement is appended at final edit, following the sudden and unexpected passing of ESA's EarthCARE Mission Scientist and this paper's principal author, Tobias Wehr. Tobias was eagerly looking forward to EarthCARE's launch, a mission to which he had



dedicated a not inconsiderable span of his career. His support for the science community, collaborative approach and enthusiasm for the mission science will not be forgotten, nor the tragedy that he will not get to see the fruition of his work.



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
