# Peer review of "The EarthCARE Mission - Science and System Overview"

_EGUsphere, 2022_

## Author Comment (AC1)

**Second Reviewer comments**

The paper readability can be improves by performing a minor revision of the document. Please find below some suggestions:

1) Abstract: it would be good to provide information on the orbit type (sun-synchronous) and local time of descending node (LTDN).

A fully dedicated section (Section 7) is allocated in Orbit parameters, do you prefer repeating the information in Abstract?

2) There are many acronyms introduced without spelling them (e.g. IPCC, CMIP6, GCM in section 2; DFA in Figure 7; MREF in Figure 10; Table 2, PBL; etc.). An appendix including the definition of the acronyms would help.

Typically no Acronyms appendix is used in AMT papers, do we implement for this Paper? All technical terms corrected/updated/explained in text (Main REFlector MREF is described in text so was not repeated in the Figure description as special request from Reviewer 1 not to duplicate the acronyms.

A list has been provided by authors for Reviewer.

The Royal Netherlands Meteorological Institute (KNMI)

Leibniz Institute for Tropospheric Research (TROPOS)

the Free University of Berlin (FUB)

McGill University,

the European Centre for Medium-Range Weather Forecasts (ECMWF)

the Royal Meteorological Institute of Belgium (RMIB)

Environment and Climate Change Canada (ECCC)

Laboratoire de météorologie dynamique (LMD) at Sorbonne University

GMV :As far as I know GMV is no longer considered an abbreviation and is now the name of the company.

3) Section 5.2, Line 293: The CPR instrument modes are introduced without explaining them. It is assumed that the modes refer to the maximum altitude sampled in the atmosphere, but it should be clearly explained. Please clarify.

Answer provided by JAXA and text updated.

4) Line 404: The solar calibration is performed over the South or North Polar Region? Unclear from the text.

South Polar region, text updated

5) Please consider swapping figure 16 and 17, since 17 is referred first in the text (line 414), followed by figure 16 (line 450). This is applicable also in other parts of the paper.

Suggestion Implemented: Figures Swapped, also figures 5-6 corrected in order of reference.

6) Figure 17 caption mentions solid horizontal grey lines, but they are not visible. Alternatively, should be solid black lines? Please check consistency between figure and caption.

Suggestion Implemented: All black lines

7) Lines 488-493: This text is potentially confusing. First, it must be noted that the section related to the products is introduced later in the paper (section 8.2), and this could already generate some confusion in a general reader. Moreover, the text refers to "not unfiltered" BBR Level 1 products. The use of a double negative in English can lead to a further confusing interpretation. It is suggested to use "artefacts-corrected" or simply "corrected" instead of "unfiltered".

Author would prefer keeping the unfiltered term as it has been used previously in many other publications. Text was altered to minimise misunderstanding.

8) Line 522: what is a PGM-HT cage?

PGH-HT is a solid (dry) lubricant, thus a Dry self lubricating cage.

9) Figure 22: The figure shows the point spread functions for the BBR TW and SW channels, but it does not clearly indicate which is which, namely red vs. violet. Please clarify the caption.

Caption updated and corrected.

**Typos/text**

Table 2: ATLID is not mentioned in this table, it is only mentioned as "the instrument"

Fixed, according to recommendation

Line 144: performaces - performances

Fixed, according to recommendation

Line 219, format of reference – remove A. G.

Fixed, according to recommendation

Line 235: suggestion - The output from this laser consists of linearly...

Fixed, according to recommendation

Line 265: th perfomace - performance

Fixed, according to recommendation.

Line 393: dedudancy  - redundancy

Fixed, according to recommendation

Line 394: channelss – channels

Fixed, according to recommendation

Line 604: Fligh - Flight

Fixed, according to recommendation

Line 701: Please add "the" before "Joint Simulator"

Fixed, according to recommendation

Line 703: Please add "the" before before "Joint Simulator" and "Satellite Data Simulator"

Fixed, according to recommendation

**Authors Response to reviewer**

Authors would like to thank the reviewer for the time spent to do this review and the acceptance of the paper.